# Universality and Limitations of Prompt Tuning

**Yihan Wang**
UCLA
`wangyihan617@gmail.com`

**Jatin Chauhan**
UCLA
`chauhanjatin100@gmail.com`

**Wei Wang**
UCLA
`weiwang@cs.ucla.edu`

**Cho-Jui Hsieh**
Google and UCLA
`chohsieh@cs.ucla.edu`

## Abstract

Despite the demonstrated empirical efficacy of prompt tuning to adapt a pretrained language model for a new task, the theoretical underpinnings of the difference between "tuning parameters before the input" against "the tuning of model weights" are limited. We thus take one of the first steps to understand the role of soft-prompt tuning for transformer-based architectures. By considering a general purpose architecture, we analyze prompt tuning from the lens of both: universal approximation and limitations with finite-depth fixed-weight pretrained transformers for continuous-valued functions. Our universality result guarantees the existence of a strong transformer with a prompt to approximate any sequence-to-sequence function in the set of Lipschitz functions. The limitations of prompt tuning for limited-depth transformers are first proved by constructing a set of datasets, that cannot be memorized by a prompt of any length for a given single encoder layer. We also provide a lower bound on the required number of tunable prompt parameters and compare the result with the number of parameters required for a low-rank update (based on LoRA) for a single-layer setting. We finally extend our analysis to multi-layer settings by providing sufficient conditions under which the transformer can at best learn datasets from invertible functions only. Our theoretical claims are also corroborated by empirical results.

## 1 Introduction

The surge in the empirical research of large-scale models has led to the emergence of a new paradigm of prompt tuning. Current large models consist of billions of parameters [Brown et al., 2020, Chowdhery et al., 2022], which greatly exacerbate the cost of tuning the entire model weights via gradient-based optimization. On the other hand, the power of scale in both model size and pretraining dataset size has demonstrated strong capabilities by achieving reasonable performance through a learnable prompt appended before the input [Li and Liang, 2021, Lester et al., 2021]. Despite this, several questions emanate around the abilities and limitations of prompt tuning.

In this work, we aim to characterize some natural yet essential questions about prompt tuning with transformer architectures. Firstly, are prompts universal approximators, i.e. with a fixed pretrained transformer network, can we find a prompt to approximate any sequence-to-sequence function in a given space? If yes, can we construct the transformer for this universality result? Second, can we identify failure modes of prompt tuning when applied on potentially non-optimal but non-trivial transformers? Moreover, since prompt tuning is usually compared against LoRA[Hu et al., 2021] in consideration to parameter-efficient tuning, is prompt tuning then more/less parameter-efficient than LoRA? Answering these questions can lead to important insights on *when* and *how* to perform prompt tuning to adapt a pretrained transformer network to a given downstream task of interest.

37th Conference on Neural Information Processing Systems (NeurIPS 2023).

In this work, we seek to answer these questions with appropriate theoretical analysis and further validate our claims with empirical results. We first characterize the universal nature of prompt tuning by constructing a specific transformer network. We show that for a given approximation error and the space of sequence-to-sequence Lipschitz functions, we can construct a transformer network, with a suitable number of layers, that can leverage prompt tuning to approximate any function in this space. Despite this universality of prompt tuning with a carefully constructed pretrained transformer, we then identify some limitations of prompt tuning with weaker but non-trivial transformers. We prove this by constructing sequence-to-sequence datasets with shared input tokens, which are surprisingly simple but cannot be memorized by prompt tuning for a given transformer. We also extend our analysis to more general settings where the shared token is not required. In this setting, we first prove that prompt tuning on a single-layer transformer requires $\Omega(n)$ trainable parameters to memorize $n$ training examples, wherein for LoRA, it suffices with $O(n)$ trainable parameters. We finally extend our analysis to the multi-layer setting and provide sufficient conditions under which prompt tuning exhibits extremely limited capacity to at best memorizing datasets from invertible functions.

Our contributions can be summarized as below:

- We characterize the universal nature of prompt tuning by explicitly constructing a transformer network (Theorem 1).
- We provide a construction-based argument for sequence-to-sequence datasets that cannot be learned by prompt tuning with a given single-layer transformer (Theorem 2).
- We provide the lower bound on the required number of parameters for prompt tuning to memorize any sequence-to-sequence functions (Theorem 3).
- We provide a sufficient condition for multi-layer transformers, under which datasets with shared output tokens cannot be learned with prompt tuning (Theorem 4).
- We conduct empirical studies, including real-world datasets, to verify our theoretical claims.

## 2   Related Work

**Theoretical Analysis of Transformers**   Various works have characterized the theoretical properties of transformers and its primary self-attention component. Yun et al. [2020] studies the universal approximation ability of transformers for continuous permutation equivariant sequence-to-sequence functions with compact support and further examined the role of using positional encodings to circumvent permutation equivariant condition. Pérez et al. [2021] show that transformer with a hard-attention is Turing complete based on their capacity to perform computations and access the internal dense representations of the data. Wei et al. [2021] further shows that transformers can approximate Turing machines with bounded computation time with a new notion of approximation. [Dong et al., 2021] provides a negative yet interesting result signifying the limitations of pure self-attention in terms of rank diminishing of the input. Other works including Kim et al. [2021], Dasoulas et al. [2021] derive upper bounds on the Lipschitz constant of respective modifications of the attention mechanism. The works by Li et al. [2022], Zhang et al. [2020] documented optimization perspective on transformer training via SGD.

**Fine-tuning and Prompt Tuning**   Fine-tuning is the standard way to adapt a pretrained model to downstream tasks. The most standard and popular paradigm is tuning the model weights via a suitable optimization procedure along with a linear head on the output representations [Radford et al., 2018, Devlin et al., 2019]. Subsequent works studied more parameter-efficient ways of fine-tuning by updating either a subset of model parameters [Ben Zaken et al., 2022] or restricting the parameter-updates to a low-dimensional subspace [Aghajanyan et al., 2021, Hu et al., 2021, Mahabadi et al., 2021]. The work by Hu et al. [2021] (their framework referred to as LoRA) has garnered particular interest in the community and Malladi et al. [2023] has provided an interpretation of LoRA via the kernel mechanism. In the particular context of LLMs, prompt tuning has emerged as the de facto approach where only the prompt is updated while keeping the rest of the transformer weights and architecture fixed [Shin et al., 2020, Lester et al., 2021, Li and Liang, 2021].

**Analysis of Prompt Tuning**   Wei et al. [2022] studies the link between prompt tuning and downstream tasks with an underlying latent variable generative model of text, which is confined to a Hidden Markov Model. However, they focused on the discrete vocabulary setting contrary to our

results for continuous sequence-to-sequence functions. Some more recent works [Akyürek et al., 2023, Von Oswald et al., 2023] characterize an intriguing property of a specific form of prompting, referred to as in-context learning, where they proved by construction that transformers can implement learning algorithms for linear models based on gradient descent and closed-form ridge regression. This work however pursued a different and specific direction from the prompting results we aim to provide for generic settings.

**Memorization Capacity of Neural Networks**   A series of works have sought to provide finite sample universal memorization capacity results of neural networks and the understanding of expressive power of neural networks. Huang and Huang [1990], Huang and Babri [1998], Huang [2003], Yamasaki [1993] analyze the memorization capacity of FNNs with sigmoid and other bounded activation functions. Hardt and Ma [2016], Zhang et al. [2021], Nguyen and Hein [2018] provide results for modern ReLU networks including FNNs and CNNs. For transformer architectures, Kim et al. prove that transformers can memorize a dataset with finite parameters. To the best of our knowledge, similar results for prompt tuning have not been studied in continuous settings for transformer architectures.

# 3   Transformers and Parameter Efficient Training

## 3.1   Preliminaries

We use the following notations throughout the paper. A bold lower case character, e.g. $\mathbf{x}$, denotes a vector. A bold upper case character, e.g. $\mathbf{W}$, denotes a matrix while $\mathbf{W}_{i,j}$, $\mathbf{W}_{i,:}$ and $\mathbf{W}_{:,j}$ is the $(i,j)$-th element, $i$-th row, $j$-th column, respectively. We use a single superscript or subscript to denote the index of a matrix, e.g. $\mathbf{X}_i, \mathbf{X}^i$ denote the $i$-th matrix in a matrices sequence. We use $\sigma$ and $\bar{\sigma}$ for softmax and hardmax operators, respectively. We use $\texttt{ReLU}(\mathbf{v}) = \max(\mathbf{v}, \mathbf{0})$ to denote the ReLU activation function where $\max(\cdot)$ function is applied entry-wise to a vector. We use $\texttt{Cone}(\mathbf{a}_1, \mathbf{a}_2, ..., \mathbf{a}_m)$ to denote a cone without its origin point where $\texttt{Cone}(\mathbf{a}_1, \mathbf{a}_2, ..., \mathbf{a}_m) = \{\mathbf{x} : \mathbf{x} = \sum_{i=1}^{m} a_i \mathbf{a}_i, a_i > 0\}$. We also define the minus operation between a set $S$ and a vector $\mathbf{v}$ as $S - \mathbf{v} = \{\mathbf{x} - \mathbf{v} : \mathbf{x} \in S\}$. In Section 4, we use $[a : b : c]$ to denote a grid $\{a, a+b, a+2b, ..., c-b\}$ from $a$ to $c$, with an interval $b$.

Transformer networks [Vaswani et al., 2017] are a stack of multiple transformer layers, composed subsequently. A transformer layer has two key components: an attention layer and a token-wise MLP layer, with residual connections around both blocks. We consider the input and output to be sequences of tokens $\mathbf{X} \in \mathbb{R}^{d \times m}$ and $\mathbf{Y} \in \mathbb{R}^{d \times m}$, where $m$ is the number of tokens in the sequence and $d$ is the token dimension.

**Definition 1** (Attention Layer). *We define an $h$-head attention layer parameterized with $\mathbf{W}_q, \mathbf{W}_k, \mathbf{W}_v, \mathbf{W}_o$ between a single token $\mathbf{x}$ and a token sequence $\mathbf{X}$ as*

$$\mathit{Att}(\mathbf{x}, \mathbf{X}) = \sum_{i=1}^{h} \mathbf{W}_o^i \mathbf{W}_v^i \mathbf{X} \cdot \sigma((\mathbf{W}_k^i \mathbf{X})^\top \mathbf{W}_q^i \mathbf{x}). \tag{1}$$

*The normalizing factor of $\frac{1}{\sqrt{d_{kq}}}$ is subsumed in the weight matrices $\mathbf{W}_k^i$ for notational simplicity.*

*We can then define the cross attention between two sequences $\mathbf{X}_1 \in \mathbb{R}^{d \times m_1}$ and $\mathbf{X}_2 \in \mathbb{R}^{d \times m_2}$ (We use $\mathbf{x}_k = (\mathbf{X}_1)_{:,k}$ for simplicity):*

$$\mathit{Att}(\mathbf{X}_1, \mathbf{X}_2) = [\mathit{Att}(\mathbf{x}_1, \mathbf{X}_2), \mathit{Att}(\mathbf{x}_2, \mathbf{X}_2), ..., \mathit{Att}(\mathbf{x}_{m_1}, \mathbf{X}_2)].$$

**Definition 2** (Standard Transformer Layer). *With definition 1, we define a standard transformer layer $\tau$ as*

$$MLP(\mathbf{X}) = [\mathbf{W}_2 ReLU(\mathbf{W}_1 \mathbf{X}_{:,1} + \mathbf{b}_1) + \mathbf{b}_2 + \mathbf{X}_{:,1}, ..., \mathbf{W}_2 ReLU(\mathbf{W}_1 \mathbf{X}_{:,n} + \mathbf{b}_1) + \mathbf{b}_2 + \mathbf{X}_{:,n}] \tag{2}$$

$$\tau(\mathbf{X}) = MLP(\mathit{Att}(\mathbf{X}, \mathbf{X}) + \mathbf{X}). \tag{3}$$

*The definition here omits the layer normalization block for simplicity (following [Kim et al., 2021]).*

We denote the set of transformer networks with $h$ heads of size $s$ and $r$ MLP hidden neurons with $\mathcal{T}^{h,s,r}$. In Section 4, we utilize a modified transformer network with hardmax operation $\bar{\sigma}$ instead of softmax $\sigma$. We denote this modified version of transformer networks as $\bar{\mathcal{T}}^{h,s,r}$.

During fine-tuning, we optimize the matrices $\mathbf{W}_q^i, \mathbf{W}_k^i, \mathbf{W}_v^i$ in the attention layer and $\mathbf{W}_1, \mathbf{W}_2, \mathbf{b}_1, \mathbf{b}_2$ in the MLP layer pertaining to a loss function $\mathcal{L}$. However in prompt tuning, the pretrained model weight matrices are fixed and we optimize a tunable sequence prepended to the input.

**Prompt Tuning** Given a pretrained transformer network $g \in \mathcal{T}$ and a downstream training dataset $S = \{(\mathbf{X}_1, \mathbf{Y}_1), ..., (\mathbf{X}_n, \mathbf{Y}_n)\}$, prompt tuning seeks to find a prompt $\mathbf{P}^* \in \mathbb{R}^{d \times m_p}$ with $m_p$ tunable tokens under the loss function $\mathcal{L}$:

$$\mathbf{P}^* = \arg\min_{\mathbf{P}} \sum_{i=1}^{n} \mathcal{L}(g([\mathbf{P}, \mathbf{X}_i])_{:,m_p:}, \mathbf{Y}_i). \tag{4}$$

The tunable prompt $\mathbf{P}$ is shared amongst all the inputs in a task. Note that $\mathbf{P}$ in prompt tuning is a continuously trainable parameter, alternately referred to as soft prompt, which is different from hard prompt in that the latter operates on a discrete space of predefined vocabulary. Since the representation power of soft prompts is strictly more than the hard prompts, the limitations studied in this paper also extend to hard prompts.

In the subsequent sections, we analyze the universality and limitations of prompt tuning while comparing the latter against fine-tuning and LoRA[Hu et al., 2021], which is a low-rank version of model fine-tuning. In Section 4, we prove that prompt tuning can be universal approximators for sequence-to-sequence functions, while providing the construction for the same. In Sections 5 and 6, we identify the failure modes where prompt tuning cannot learn with a possibly non-optimal but non-trivial pretrained transformer network.

## 4 Universality of Prompt Tuning

Without loss of generality, we assume that the support and range set of all considered sequence-to-sequence functions $f$ is $[0, 1]^{d \times m}$ in this section. We define $\mathcal{F}_L$ as the collection of all continuous sequence-to-sequence $L$-lipschitz functions under norm $p$ and sequence length $m$. For $f \in F_L$ and any two inputs $\mathbf{X}, \mathbf{X}' \in [0, 1]^{d \times m}$, we have $\|f(\mathbf{X}) - f(\mathbf{X}')\|_p \leq L\|\mathbf{X} - \mathbf{X}'\|_p$. Furthermore, given functions $f_1, f_2$, the approximation error under a $p$-norm (which is entry-wise) is measured as:

$$d_p(f_1, f_2) = (\int \|f_1(\mathbf{X}) - f_2(\mathbf{X})\|_p^p d\mathbf{X})^{\frac{1}{p}}. \tag{5}$$

Primarily, we show that there exists a Transformer network $g \in \mathcal{T}^{2,1,4}$ such that for any $f \in \mathcal{F}_L$, prompt tuning on $g$ can approximate this function upto some error budget $\epsilon > 0$.

**Theorem 1.** *Let $1 \leq p < \infty$ and $\epsilon > 0$, there exist a transformer network $g \in \mathcal{T}^{2,1,4}$ and prompt length $m_p$, such that for any $f \in \mathcal{F}_L$ we can find a prompt $\mathbf{P} \in \mathbb{R}^{d \times m_p}$ with $d_p(g([\mathbf{P}, \cdot])_{:,m_p:}, f) \leq \epsilon$.*

Here we use the transformer in a encoder mode which generates the $m$ outputs in one step. In Appendix C.4, a similar result can be obtained for next-token prediction, which is widely used in many recent language models.

The proof is inspired from [Yun et al., 2019a], which follows the typical construction based proof mechanism to show universality. Thereby, we can construct a "meta-transformer" for prompt tuning to approximate any sequence-to-sequence function with prompt tuning. Next we briefly describe the two steps for the construction of this meta-transformer. We start by building a meta-function for $\mathcal{F}_L$.

**Building the Meta-Function** We denote the length of all inputs as $m$ and the prompt length as $m_p$. Then we can build a sequence-to-sequence meta-function that accepts inputs with length $m + m_p$.

**Lemma 1.** *For the sequence-to-sequence function space $\mathcal{F}_L$ with functions $f : [0, 1]^{d \times m} \rightarrow [0, 1]^{d \times m}$, we can build a sequence-to-sequence function $\bar{g} : [0, 1]^{d \times (m_p+m)} \rightarrow [0, 1]^{d \times (m_p+m)}$ such that for any $f \in \mathcal{F}_L$, we can find $\mathbf{P} \in \mathbb{R}^{d \times m_p}$, $d_p(\bar{g}([\mathbf{P}, \cdot])_{:,m_p:}, f) \leq \epsilon/2$.*

The complete proof is given in Appendix C.1. Succinctly, we first quantize the input and output sequence space of $[0, 1]^{d \times m}$ into a grid $G_{\delta,m} = \{0, \delta, 2\delta, ..., 1 - \delta\}^{d \times m}$, thus leading to $C = (\frac{1}{\delta^{d \times m}})^{\frac{1}{\delta^{d \times m}}}$ possible functions mappings from the input to the output, in this discrete space. By

this quantized function space as $\bar{\mathcal{F}}_L = \{\bar{f}_1, \bar{f}_2, ..., \bar{f}_C\}$, we can select $\delta$ such that the approximation error for any function is less than $\epsilon/2$. Then we construct a set of quantized prompts in $G_{\delta,m_p} = \{0, \delta, 2\delta, ..., 1 - \delta\}^{d \times m_p}$ to index these $C$ functions and construct a quantized function $\bar{g}$ where $\bar{g}([\mathbf{P}_i, \mathbf{X}])_{:,m_p:} = \bar{f}_i(\mathbf{X}), i = 1, 2, ..., C$, for all $\mathbf{X} \in G_{\delta,m}$, thereby concluding the lemma.

Next we can utilize some conclusions in [Yun et al., 2019a] to construct a transformer for $\bar{g}$.

**Constructing the Meta-Transformer** We first introduce a useful lemma which enables the construction of a transformer for any quantized sequence-to-sequence function.

**Lemma 2.** *For any given quantized function* $\bar{f} : [0, 1]^{d \times m} \to [0, 1]^{d \times m}$ *with quantization at interval* $\delta$, $\exists \bar{h} \in \bar{\mathcal{T}}^{2,1,1}$ *such that* $\bar{f} = \bar{h}$ *with positional embedding* $\mathbf{E} = \begin{bmatrix} 0 & 1 & 2 & ... & m-1 \\ 0 & 1 & 2 & ... & m-1 \\ \vdots & \vdots & \vdots & \ddots & \vdots \\ 0 & 1 & 2 & ... & m-1 \end{bmatrix}$.

The proof mainly follows the discussions in Section C of [Yun et al., 2019a]. To prove this lemma, the network $\bar{h}$ can be constructed in the following three steps. We first use a series of MLP layers to quantize the input to grid $[0 : \delta : 1 - \delta]^{d \times m}$ and then a series of attention layers to obtain a unique contextual mapping for each quantized input. Finally we can use a series of MLP layers to map the unique contextual mapping to the desired outputs. While a transformer network usually stacks self-attention and MLP layers alternately within a single layer, the aforementioned construction can be trivially attained via the use of skip connections. The complete proof of Lemma 2 is deferred to Appendix C.2.

Since $\bar{g}$ is a quantized function in grid $G_{\delta,m+m_p}$, following Lemma 2 we can find a modified version of transformer $\bar{h} \in \bar{\mathcal{T}}^{2,1,1}$ such that $\bar{g}([\mathbf{P}, \mathbf{X}]) = \bar{h}([\mathbf{P}, \mathbf{X}])$. The modified version of transformer $\bar{g}$ with hardmax operators can then be approximated with a standard transformer $g$ with softmax operators by Lemma 3.

**Lemma 3** (Lemma 9 in [Yun et al., 2019a]). *For each* $\bar{h} \in \bar{\mathcal{T}}^{2,1,1}$, $\epsilon > 0$ *and* $1 \leq p < \infty$, $\exists g \in \mathcal{T}^{2,1,4}$ *such that* $d_p(\bar{h}, g) \leq \epsilon/2$.

Since the approximation error can be treated uniformly amongst the $\mathbf{P}_i$, we have that $d_p(\bar{h}([\mathbf{P}_i, \cdot])_{:,m_p:}, g([\mathbf{P}_i, \cdot])_{:,m_p:}) \leq d_p(\bar{h}([\mathbf{P}_i, \cdot]), g([\mathbf{P}_i, \cdot])) \leq \epsilon/2$. Therefore, we can build a transformer $g \in \mathcal{T}^{2,1,4}$, such that for any sequence-to-sequence $f \in \mathcal{F}_L$, we can find a quantized version $\bar{f}_i \in \bar{\mathcal{F}}_L$ and the corresponding prompt $\mathbf{P}_i \in G_{\delta,m_p}$ such that

$$d_p(g([\mathbf{P}_i, \cdot])_{:,m_p:}, f) \leq d_p(g([\mathbf{P}_i, \cdot])_{:,m_p:}, \bar{h}([\mathbf{P}_i, \cdot])) + d_p(\bar{h}([\mathbf{P}_i, \cdot])_{:,m_p:}, \bar{f}_i) + d_p(\bar{f}_i, f) \leq \epsilon. \quad (6)$$

Theorem 1 provides the construction for a large transformer (discussed more in appendix) that is sufficient for prompt tuning to exhibit universal approximation over a Lipschitz function space. However, even this strong transformer also has limitations with prompt tuning when the target function $f \notin \mathcal{F}_L$. Is this an essential limitation for prompt tuning on any transformer? In the next section, we will theoretically analyze the limitations of prompt tuning with transformers and target functions under more general conditions.

## 5 Limitations of Prompt-Tuning: Single Layer Transformer

To analyse the failure modes and therefore the limitations under the setting where a transformer has fixed pretrained weights, we follow the lens of exact memorization in the subsequent sections.

**Definition 3** (Memorization of a Sequence-to-Sequence Dataset). *Given a sequence-to-sequence dataset* $S = \{(\mathbf{X}_1, \mathbf{Y}_1), ..., (\mathbf{X}_n, \mathbf{Y}_n)\}$ *where* $\mathbf{X}_i, \mathbf{Y}_i \in \mathbb{R}^{d \times m}$ *are the input/output sequences, we consider a function* $f$ *exactly memorizing dataset* $S$ *if* $f(\mathbf{X}_i) = \mathbf{Y}_i$. *In the following proofs of this section, we explicitly focus on the last output token, ie:* $f(\mathbf{X}_i)_{:,-1} = (\mathbf{Y}_i)_{:,-1}$.

We start from the analysis on a single layer transformer and extend to multi-layer settings in Section 6.

## 5.1 Failure modes of Prompt Tuning

It is straightforward to note that prompt tuning has limited expressive power when the number of trainable parameters is limited. A natural question to then ask is: Does increasing the number of trainable prompt tokens suffice? While it is known that for MLPs, even with a single hidden layer, increasing the number of hidden neurons can memorize any training data [Yun et al., 2019b]. However, as we will prove next, this is not the case for prompt tuning. This result highlights an essential limitation of prompt tuning compared to model fine-tuning.

Before providing the theorem statement, we first outline some straightforward assumptions on the pretrained transformer and datasets, without which prompt tuning trivial loses expressive power.

We consider sequence-to-sequence datasets of the form $S = \{(\mathbf{X}_1, \mathbf{Y}_1), (\mathbf{X}_2, \mathbf{Y}_2), ..., (\mathbf{X}_n, \mathbf{Y}_n)\}$ with $n$ distinct examples and a single-layer single-head standard transformer defined in Definition 2. The results can be directly extended to the single-layer multi-head scenario, which we skip here to avoid notational clutter.

**Assumption 1** (Non-trivial conditions). *We assume that all output tokens $(\mathbf{Y}_i)_{:,k}$ are in the range set of MLP, otherwise the expressivity becomes trivially weak. We assume that $\mathbf{W}_q, \mathbf{W}_k, \mathbf{W}_v$ are full rank matrices and that $\mathtt{Att}(\mathbf{X}_i, \mathbf{X}_i) + \mathbf{X}_i$ are distinct for $i = 1, 2, ..., n$.*

**Assumption 2** (Assumption for the MLP layer). *We assume that $d \geq 2 + \mathit{dim}((\mathtt{MLP}^{-1}(\mathbf{y}_{10}) - \mathbf{x}_0) \cup (\mathtt{MLP}^{-1}(\mathbf{y}_{20}) - \mathbf{x}_0))$ for the dataset constructed in Theorem 2 and token dimension $d$. $\mathit{dim}(\mathcal{S})$ measures the dimension of subspace spanned by vectors in a set $\mathcal{S}$ and $\mathtt{MLP}^{-1}(\mathbf{y}) = \{\mathbf{x} : \mathtt{MLP}(\mathbf{x}) = \mathbf{y}\}$.*

*We provide an example for this assumption in Example 1 and a sufficient condition in the following Lemma 4.*

**Lemma 4.** *If $\|\mathbf{W}_1\|_2 \times \|\mathbf{W}_2\|_2 < 1$, where $\|\cdot\|_2$ is the matrix spectral norm, then the MLP block in Definition 2 is invertible, ie, $\mathtt{MLP}^{-1}$ is a singleton set.*

*Therefore, if Lemma 4 holds and $d \geq 4$, Assumption 2 also holds.*

Proof of Lemma 4 can be found in Appendix C.5. The experimental evidence in [Dong et al., 2021] shows that for most architectures, the norm of the weight matrices indeed admits small values and thus the requirement that $\|\mathbf{W}_1\|_2 \times \|\mathbf{W}_2\|_2 < 1$ is a mild condition.

With these assumptions, here we introduce our first theorem on the unlearnability of prompt tuning.

**Theorem 2.** *For a single layer transformer $\tau$ defined above with Assumptions 1 and 2, we can build a sequence-to-sequence dataset $S = \{(\mathbf{X}_1 = [\mathbf{x}_1, \mathbf{x}_0], \mathbf{Y}_1 = [\mathbf{y}_{11}, \mathbf{y}_{10}]), (\mathbf{X}_2 = [\mathbf{x}_2, \mathbf{x}_0], \mathbf{Y}_2 = [\mathbf{y}_{21}, \mathbf{y}_{20}]))\}$, and we cannot find a prompt $\mathbf{P} \in \mathbb{R}^{d \times m_p}$ with any $m_p > 0$ such that $\tau([\mathbf{P}, \mathbf{X}_i]) = \mathbf{Y}_i$ holds for any $i = 1, 2$. The vectors $\mathbf{x}_0, \mathbf{x}_1, \mathbf{x}_2$ are denoted post positional encodings.*

An important feature of this dataset is that the same token $\mathbf{x}_0$ is shared between the two examples, and the expressive capability of prompt tuning is limited by the correlation of outputs corresponding to this token in different examples. We show a concrete example here to illustrate this theorem (note that Lemma 4 is in fact not required in the following construction) and defer the formal proof to Appendix C.6.

**Example 1.** *We consider a single-head transformer layer $\tau$, where $\mathbf{b}_1 = \mathbf{b}_2 = \mathbf{0}$, $\mathbf{W}_1 = 1^{r \times d}$, $\mathbf{W}_2 = 1^{d \times r}$. Then the token-wise MLP layer is a concatenation of two linear functions:*

$$\mathtt{MLP}(\mathbf{x}) = \begin{cases} (\mathbf{W}_2\mathbf{W}_1 + \mathbf{I})\mathbf{x}, & (\mathbf{W}_1\mathbf{x})_0 > 0 \\ \mathbf{x} & , (\mathbf{W}_1\mathbf{x})_0 \leq 0 \end{cases} \tag{7}$$

*Here $(\mathbf{W}_1\mathbf{x})_0$ denotes the first element of vector $\mathbf{W}_1\mathbf{x}$.*

$\mathbf{W}_2\mathbf{W}_1 + \mathbf{I}$ is a non-singular matrix. Therefore, for any $\mathbf{y}$ in $\mathtt{MLP}(\mathbf{X})$'s output set, $\mathtt{MLP}^{-1}(\mathbf{y})$ contains at most two points $\{\mathbf{y}, (\mathbf{W}_2\mathbf{W}_1 + \mathbf{I})^{-1}\mathbf{y}\}$. We arbitrarily choose $\mathbf{x}_0, \mathbf{y}_{10}$ and $\mathbf{y}_{20}$.

As long as $d \geq 6$ (from Assumption 2), we can find $\mathbf{c}_1, \mathbf{c}_2$ such that $\mathbf{c}_1, \mathbf{c}_2 \perp \mathbf{y}_{10} - \mathbf{x}_0, \mathbf{y}_{20} - \mathbf{x}_0, (\mathbf{W}_2\mathbf{W}_1 + \mathbf{I})^{-1}\mathbf{y}_{10} - \mathbf{x}_0, (\mathbf{W}_2\mathbf{W}_1 + \mathbf{I})^{-1}\mathbf{y}_{20} - \mathbf{x}_0, \mathbf{c}_1 \perp \mathbf{c}_2$. Then we choose $\mathbf{x}_1$ and $\mathbf{x}_2$ such that $\mathtt{Att}(\mathbf{x}_0, \mathbf{X}_1) \parallel \mathbf{c}_1$ and $\mathtt{Att}(\mathbf{x}_0, \mathbf{X}_2) \parallel \mathbf{c}_2$ (Lemma 7 in Appendix). Then $\mathtt{Cone}(-\mathtt{Att}(\mathbf{x}_0, \mathbf{X}_1), \mathbf{a} - \mathbf{x}_0) \cap \mathtt{Cone}(-\mathtt{Att}(\mathbf{x}_0, \mathbf{X}_2), \mathbf{b} - \mathbf{x}_0) = \emptyset$, for any $\mathbf{a} \in \{\mathbf{y}_{10}, (\mathbf{W}_2\mathbf{W}_1 + \mathbf{I})^{-1}\mathbf{y}_{10}\}$ and $\mathbf{b} \in \{\mathbf{y}_{20}, (\mathbf{W}_2\mathbf{W}_1 + \mathbf{I})^{-1}\mathbf{y}_{20}\}$. Here $\mathtt{Cone}$ stands for a convex cone as defined in Section 3.1.

If a $\mathbf{P}$ exists such that $\tau([\mathbf{P}, \mathbf{X}_i]) = \mathbf{Y}_i$ holds for both $i = 1, 2$, then we have

$$\mathtt{Att}(\mathbf{x}_0, [\mathbf{P}, \mathbf{X}_1]) = \lambda(\mathbf{X}_1, \mathbf{x}_0, [\mathbf{P}, \mathbf{X}_1])\mathtt{Att}(\mathbf{x}_0, \mathbf{X}_1) + \lambda(\mathbf{P}, \mathbf{x}_0, [\mathbf{P}, \mathbf{X}_1])\mathtt{Att}(\mathbf{x}_0, \mathbf{P}) \quad (8)$$
$$\mathtt{Att}(\mathbf{x}_0, [\mathbf{P}, \mathbf{X}_2]) = \lambda(\mathbf{X}_2, \mathbf{x}_0, [\mathbf{P}, \mathbf{X}_2])\mathtt{Att}(\mathbf{x}_0, \mathbf{X}_2) + \lambda(\mathbf{P}, \mathbf{x}_0, [\mathbf{P}, \mathbf{X}_2])\mathtt{Att}(\mathbf{x}_0, \mathbf{P})$$

where $\lambda(\cdot, \cdot, \cdot)$ is a positive scalar. We also have

$$\mathtt{Att}(\mathbf{x}_0, [\mathbf{P}, \mathbf{X}_1]) + \mathbf{x}_0 \in \mathtt{MLP}^{-1}(\mathbf{y}_{10})$$
$$\mathtt{Att}(\mathbf{x}_0, [\mathbf{P}, \mathbf{X}_2]) + \mathbf{x}_0 \in \mathtt{MLP}^{-1}(\mathbf{y}_{20})$$

as $\mathtt{MLP}(\mathtt{Att}(\mathbf{x}_0, [\mathbf{P}, \mathbf{X}_i]) + \mathbf{x}_0) = \mathbf{y}_{i0}, i = 1, 2$.

Therefore, $\mathtt{Att}(\mathbf{x}_0, \mathbf{P})$ must be in both $\mathtt{Cone}(\mathbf{a} - \mathbf{x}_0, -\mathtt{Att}(\mathbf{x}_0, \mathbf{X}_1))$ and $\mathtt{Cone}(\mathbf{b} - \mathbf{x}_0, -\mathtt{Att}(\mathbf{x}_0, \mathbf{X}_2))$, where $\mathbf{a} \in \{\mathbf{y}_{10}, (\mathbf{W}_2\mathbf{W}_1 + \mathbf{I})^{-1}\mathbf{y}_{10}\}$ and $\mathbf{b} \in \{\mathbf{y}_{20}, (\mathbf{W}_2\mathbf{W}_1 + \mathbf{I})^{-1}\mathbf{y}_{20}\}$, which contradicts the existence of $\mathbf{P}$ as $\mathtt{Cone}(-\mathtt{Att}(\mathbf{x}_0, \mathbf{X}_1), \mathbf{a} - \mathbf{x}_0) \cap \mathtt{Cone}(-\mathtt{Att}(\mathbf{x}_0, \mathbf{X}_2), \mathbf{b} - \mathbf{x}_0) = \emptyset$. Therefore, in this example, even though we allow an arbitrary number of trainable parameters in prompt $\mathbf{P}$, we cannot find one to exactly memorize the training set with only two training examples.

This theorem reveals an important difference between prompt tuning and adjusting the model weights directly. For any training dataset $S$ with two training examples $\{(\mathbf{X}_1, \mathbf{Y}_1), (\mathbf{X}_2, \mathbf{Y}_2)\}$, so long as $\mathtt{Att}(\mathbf{X}_1, \mathbf{X}_1) + \mathbf{X}_1$ and $\mathtt{Att}(\mathbf{X}_2, \mathbf{X}_2) + \mathbf{X}_2$ are distinct, MLP can easily map the post-attention features to expected output tokens with finite number of hidden neurons. As a result, tuning the MLP parameters for this pretrained transformers can memorize any dataset in the form of Assumption 1. However, prompt tuning cannot achieve this even if the number of tunable tokens $\to$ infinity, thereby limiting the expressiveness of prompt tuning when compared to model fine-tuning.

## 5.2 Comparison with a More General Dataset

In Section 5.1, we constructed sequence-to-sequence datasets that cannot be learned by a given transformer layer with prompt tuning, by utilizing the shared token between different training examples. In this section, we compare the expressive power of prompt tuning and fine-tuning under a more general dataset construction where the former requirement can be relaxed.

Since the primary essence of prompt tuning is to perform parameter-efficient tuning, wherein we seek to adapt a pretrained large model to a new task with fewer tunable parameters, we compare prompt tuning with another parameter-efficient version of model-tuning: LoRA [Hu et al., 2021]. Succinctly, we compare the required number of parameters to memorize a given dataset. Again, consider a sequence-to-sequence dataset $S = \{(\mathbf{X}_1, \mathbf{Y}_1), (\mathbf{X}_2, \mathbf{Y}_2), ..., (\mathbf{X}_n, \mathbf{Y}_n)\}$, where $\mathbf{X}_i = [\mathbf{x}_{i1}, \mathbf{x}_{i2}, ..., \mathbf{x}_{im}]$ and $\mathbf{Y}_i = [\mathbf{y}_{i1}, \mathbf{y}_{i2}, ..., \mathbf{y}_{im}]$. We again discuss the memorization of the last output token for simplicity and results can be directly extended.

We first give the required number of parameters of LoRA to memorize dataset $S$.

**Lemma 5** (LoRA). *For a standard single-layer transformer $\tau$ defined in Definition 2 with $r \geq n$ MLP hidden neurons, for any sequence-to-sequence dataset $S$ satisfying Assumptions 1, we can apply a low-rank update to MLP weights with $O(nd)$ parameters to memorize $\tau(\mathbf{X}_i)_{:,m} = \mathbf{y}_{im}$.*

This lemma is derived based on the memorization capabilities of 1-hidden layer MLPs [Yun et al., 2019b]. As the post-attention values for different training inputs are different from Assumption 1, we can construct a low rank update with $O(nd)$ parameters on the MLP layer to memorize $S$. We defer the complete proof to Appendix C.7.

For prompt tuning, we derive a result in the next theorem which shows that it requires $\Omega(nd)$ tunable parameters to memorize some constructed dataset $S$ with $n$ examples.

**Theorem 3** (Lower bound on Tunable Prompt Parameters). *For any single layer transformer $\tau$ defined in Definition 2, there exists a sequence-to-sequence dataset $\{(\mathbf{X}_1 = [\mathbf{x}_{10}, \mathbf{x}_1], [\mathbf{y}_{10}, \mathbf{y}_{11}]), (\mathbf{X}_2 = [\mathbf{x}_{20}, \mathbf{x}_2], [\mathbf{y}_{20}, \mathbf{y}_{21}]), ..., (\mathbf{X}_n = [\mathbf{x}_{n0}, \mathbf{x}_n], [\mathbf{y}_{n0}, \mathbf{y}_{n1}])\}$ that satisfies Assumption 1 with $n < d$ training examples such that we need at least $n$ prompt tokens in $\mathbf{P}$ to memorize the training set, ie, for $\tau([\mathbf{P}, \mathbf{X}_i])_{:,-1} = \mathbf{y}_{i1}$ to hold for all $i = 1, 2, ..., n$.*

This dataset can be constructed by including $n$ examples that require $n$ linearly independent prompts tokens. The complete proof is deferred to Appendix C.8.

Note that in Theorem 3, we provide a key lower bound on the required number of prompt tokens for exact memorization and this can very well more than $nd$. This partially (but not necessarily) explains the worse empirical performance of prompt tuning against LoRA under a comparable number of trainable parameters.

# 6   Extension to Multi-Layer Setting

In this section, we extend our analysis to multi-layer setting and provide a sufficient condition under which the expressiveness of prompt tuning is restricted. An immediate consequence of our result is an interesting connection to the spectral norm of soft prompts surfaces. This result provides us a partial understanding of the phenomenon that soft prompt $\mathbf{P}$ vectors typically exhibit larger norms compared to the actual input $\mathbf{X}$, after the tuning.

With some further notation adjustments, we denote an $H$ layer pretrained transformer network as $g(\in \mathcal{T}) = \tau^1 \circ \tau^2 \circ ... \circ \tau^H$, the input set as $\mathcal{X}^1$, and the set of possible prompts as $\mathcal{P}^1$. We assume that the following compactness condition is satisfied:

$$\|[\mathbf{P}^l, \mathbf{X}^l]\|_2 \leq D^l \tag{9}$$
$$\text{s.t. } [\mathbf{P}^{l+1}, \mathbf{X}^{l+1}] = \tau^l([\mathbf{P}^l, \mathbf{X}^l]), \forall l = 1, ..., H.$$

Here $[\mathbf{P}^1, \mathbf{X}^1]$ is the input to the first layer $\tau^1$ with $\mathbf{P}^1 \in \mathcal{P}^1$, $\mathbf{X}^1 \in \mathcal{X}^1$ and $\|\cdot\|_2$ is the spectral norm. Similarly, $[\mathcal{P}^{H+1}, \mathcal{X}^{H+1}]$ denotes the output set.

We start by providing an upper bound to the Lipschitz constant of attention, pertaining to eq 9. This derivation is different from the works of [Dasoulas et al., 2021, Vuckovic et al., 2020] and thus can be of independent interest.

**Lemma 6.** *Under the compactness condition, the Lipschitz constant of the $i$-th attention head in the $l$-th transformer layer, denoted for simplicity as $\mathit{Att}^{i,l}$, admits the following bound w.r.t the entire input sequence of length $m$:*

$$Lip(\mathit{Att}^{i,l}(\cdot, \cdot)) \leq (1 + 8\sqrt{m}(D^l)^2 \|(\mathbf{W}_k^{i,l})^T \mathbf{W}_q^{i,l}\|_2) \|\mathbf{W}_v^{i,l}\|_2, \tag{10}$$

*and the Lipschitz constant of the entire attention block in layer $l$, denoted as $\mathit{Att}^l$, admits the bound:*

$$Lip(\mathit{Att}^l(\cdot, \cdot)) \leq \sqrt{\sum_{i=1}^{h} (\|\mathbf{W}_o^{i,l}\|_2 \times Lip(\mathit{Att}^{i,l}))^2}. \tag{11}$$

It is noteworthy that this upper bound is dependent on $D^l$, the spectral norm of the input prepended with the prompt. In conjunction with the following theorem, we obtain a result on limited expressivity of prompt tuning by showing that the transformer becomes invertible, in consideration to functions from $\mathcal{P}^1 \times \mathcal{X}^1 \rightarrow \mathcal{P}^{H+1} \times \mathcal{X}^{H+1}$ (an extension to functions of the from $\mathcal{X}^1 \rightarrow \mathcal{X}^{H+1}$ is provided in Appendix Section C.11).

**Theorem 4.** *A transformer $g \in \mathcal{T}$ is invertible, ie ,$g^{-1}(\mathbf{Y}) = \{\mathbf{X} : g(\mathbf{X}) = \mathbf{Y}\}$ is a singleton set $\forall \mathbf{Y}$ in range of $g$, if:*

1. *The Lipschitz constant of the attention block in each layer $\tau^l$ is strictly less than 1*

2. *The Lipschitz constant of the 2-layer ReLU block in each layer $\tau^l$, which is bounded by $\|\mathbf{W}_2^l\|_2 \times \|\mathbf{W}_1^l\|_2$, is strictly less than 1.*

Proof of Theorem 4 can be found in Appendix C.9. Combining Lemma 6 and Theorem 4, we observe that the invertibility is guaranteed if the upper bound for the Lipschitz constant of the attention, eq 11, and the MLP layer, is strictly less than 1. In this case, we can then construct arbitrarily many datasets where two different inputs share the same output, and prompt tuning cannot learn (more subtly: memorize) these datasets with a restricted prompt norm.

# 7 Experiments

## 7.1 Experimental Settings

In Section 7.2, we use a standard single-layer single-head transformer from Definition 2, to justify the infinite prompt-length limitation. In Section 7.3, we justify the increasing prompt norm on the pretrained LLaMA 7B model [Touvron et al., 2023]. For prompt tuning and LoRA, we use the Huggingface Peft library [Mangrulkar et al., 2022]. On the dataset front, we utilize the RTE subtask of SuperGlue dataset [Wang et al., 2019] and WMT14 En-Fr translation [Bojar et al., 2014]. More details and hyperparameter settings can be found in Appendix A.

## 7.2 Limited Expressivity of Infinite Length Prompt

We first construct the dataset following the proof of Theorem 2 and then show that prompt tuning cannot memorize this simple dataset $\{(\mathbf{X}_1 = [\mathbf{x}_1, \mathbf{x}_0], \mathbf{Y}_1 = [\mathbf{y}_{11}, \mathbf{y}_{10}]), (\mathbf{X}_2 = [\mathbf{x}_2, \mathbf{x}_0], \mathbf{Y}_2 = [\mathbf{y}_{21}, \mathbf{y}_{20}])\}$ even with very large prompt lengths.

We set the token dimension $d = 10$. We follow the default pytorch weight initialization and then normalize $\mathbf{W}_1, \mathbf{W}_2$ such that $\|\mathbf{W}_2\|_2 \times \|\mathbf{W}_1\|_2 < 1$, following Assumption 2. We randomly sample $\mathbf{x}_0, \mathbf{y}_{10}, \mathbf{y}_{20}$ in a uniform distribution in $[0, 1)^d$ and construct the corresponding vectors: $\mathbf{x}_1$ and $\mathbf{x}_2$ following Theorem 2. To compute $\texttt{MLP}^{-1}(\mathbf{y})$, we follow [Kim et al., 2021] Section 4.1 with 5000 iterations at convergence. We solve $\texttt{Att}(\mathbf{x}_0, [\mathbf{x}_0, \mathbf{x}_1]) \parallel \mathbf{c}$ in Lemma 7 with gradient descent terminating at $\angle(\texttt{Att}(\mathbf{x}_0, [\mathbf{x}_0, \mathbf{x}_1]), \mathbf{c}) < 0.0001$. We repeat this setup to obtain 3 different datasets for distinct $\mathbf{x}_0, \mathbf{y}_{10}, \mathbf{y}_{20}$ and denote these with $S_i, i = 1, 2, 3$.

We perform prompt tuning, MLP fine-tuning and MLP LoRA training on the constructed datasets for 5 runs and report the mean and standard deviation of per-element Mean Squared Error (MSE) loss $\mathbf{y}_{10}, \mathbf{y}_{20}$ at convergence. We show the comparison between prompt-tuning and MLP fine-tuning in Figure 1. As we can observe from the figure, increasing the number of soft prompt tokens post a certain threshold that does not exhibit any reduction in MSE. On the contrary, fine-tuning on the MLP layer tend to easily memorize the training set by reducing the training loss to almost zero (all the three curves for fine-tuning overlap and thus not differentiated). Note that we plot the standard deviation, however it is negligible in the range. Similar to fine-tuning on the MLP layer, LoRA with width 2 on the MLP layer also achieves near-zero training loss which is less than $10^{-10}$ on the constructed dataset. We don't plot the comparison on Figure 1 as all the six curves are overlapped). This result validates our Theorem 3 that LoRA can memorize a dataset with $n$ examples with trainable parameters $O(n)$ while prompt-tuning may require more.

## 7.3 Increasing Prompt Spectral Norm during Tuning

As discussed in Section 6, a major constraint on the expressive power of prompt tuning is the spectral norm of soft prompts. In Figure 2, we plot the curve for spectral norm of soft prompt as training progresses and the loss reduces on RTE dataset. The curve for WMT14 En-Fr dataset can be found in Appendix B. This trend clearly highlights that in order to counter the limit on the capacity, the spectral norm consistently increases till the training loss saturates.

# 8 Conclusions

In this work, we embark on exploring the capabilities of prompt tuning in the continuous regime, contrasting it with fine-tuning, as an initial endeavor towards a theoretical comprehension. We prove by construction that prompt tuning admits universal approximation within the space of Lipschitz functions. Additionally, we identified inherent limitations of prompt tuning on single-layer transformers by constructing theoretically difficult datasets for prompt tuning. These limitations are then extended to multi-layer setting under a specific prompt-norm restriction.

From the analysis in Theorem 2 and 3, we note that the limitation of prompt-tuning primarily arises from the correlation across different inputs. Broadly describing, prompt-tuning implements transformation on different inputs via "additional attention values", which is more restrictive as compared to the transformations from MLP layers on input tokens. An interesting potential direction to improve prompt-tuning is: "designing a mechanism to leverage prompting in order to generate

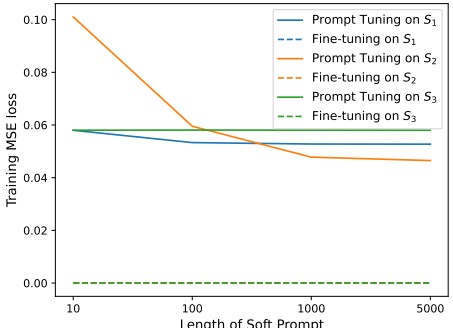
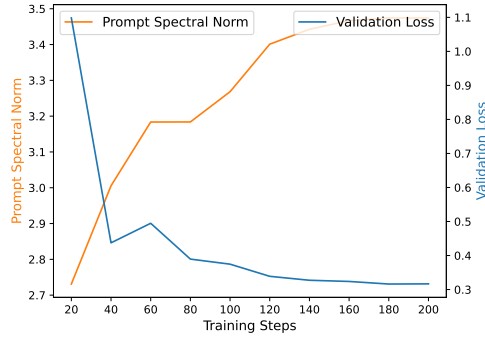

Figure 1: MSE losses at convergence for the 3 constructed datasets (following Theorem 2). We plot the bold curves with increasing prompt length in prompt tuning and dashed fixed lines in fine-tuning (all three datasets overlapping).

Figure 2: Increasing prompt spectral norm during tuning on SuperGlue RTE dataset.

prompt-dependent adapter/LoRA updates". We expect to have some future work focusing on designing novel prompt-tuning strategies along this direction.

**Limitations** While our results provide valuable insights, extending the construction in Theorem 2 to multiple layers and deriving tighter bounds for Lemma 6 are critical steps for a deeper understanding of the limitations of prompt tuning.

## Acknowledgments and Disclosure of Funding

We thank the reviewers for their invaluable feedbacks. The work is supported in part by NSF 2008173, 2048280, 2325121, 2331966, ONR N00014-23-1-2300:P00001.

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

## A Experimental Details

All the experiments are run on a NVIDIA RTX A6000 GPU. For experiments with Llama 7B model, we use batch size 32 and learning rate 0.001. For experiment on WMT14 En-Fr translation, we only compute the loss on the first 100 examples for computational efficiency.

We use Adam optimizer and optimal learning rate from grid search at 0.1 for prompt-tuning and at 0.001 for fine-tuning in Section 7.2.

In Section 7.3, we use the default loss function in Huggingface implementation for causal language models. We use prompt length $m = 10$ and the prompt tokens are initialized as the first $m$ tokens in the model vocabulary.

## B Additional Experiments

As mentioned in Section 7.3, the second real world dataset used in our experiment is WMT14 En-Fr translation in order to illustrate that the spectral norm of soft prompts increases during training. We show the curve in Figure B.

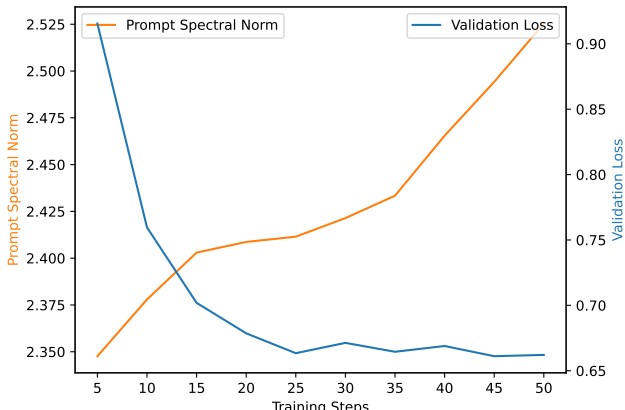

Figure 3: Increasing prompt spectral norm during tuning on WMT14 En-Fr translation dataset.

## C Proof of Lemmas and Theorems

### C.1 Proof of Lemma 1

For the sequence-to-sequence function space $\mathcal{F}_L$ with functions $f : [0,1]^{d \times m} \to [0,1]^{d \times m}$, we can build a sequence-to-sequence function $\bar{g} : [0,1]^{d \times (m_p + m)} \to [0,1]^{d \times (m_p + m)}$ such that for any $f \in \mathcal{F}_L$, we can find $\mathbf{P} \in \mathbb{R}^{d \times m_p}$, $d_p(\bar{g}([\mathbf{P}, \cdot])_{:,m_p:}, f) \leq \epsilon/2$.

*Proof.* we first quantize the input and output sequence space of $[0,1]^{d \times m}$ into a grid space $G_{\delta,m} = \{0, \delta, 2\delta, ..., 1 - \delta\}^{d \times m}$, which leads to $C = (\frac{1}{\delta^{d \times m}})^{\frac{1}{\delta^{d \times m}}}$ functions considering all input and output mappings in this grid. We index these $C$ functions as $\bar{\mathcal{F}}_L = \{\bar{f}_1, \bar{f}_2, ..., \bar{f}_C\}$. For $\mathbf{X} \notin G_{\delta,m}$, we let $\bar{f}_i(\mathbf{X}) = \bar{f}_i(\mathbf{X}^*)$ if $k_{i,j}\delta < \mathbf{X}_{i,j}, \mathbf{X}^*_{i,j} \leq (k_{i,j} + 1)\delta$ and $\mathbf{X}^* \in G_{\delta,m}$.

Then for any $f \in \mathcal{F}_L$, we can find a function $\bar{f} \in \bar{F}_L$ such that $d_p(\bar{f}, f) = (\int \|\bar{f}(\mathbf{X}) - f(\mathbf{X})\|_p^p d\mathbf{X})^{1/p} \leq (\int L^p m d \delta^p d\mathbf{X})^{1/p} = L(md)^{\frac{1}{p}}\delta$. We choose $\delta = \delta_1$ such that $L(md)^{\frac{1}{p}}\delta \leq \epsilon/2$. For the prompt part, we choose $m_p$ such that $\frac{1}{\delta^{d \times m_p}} \geq C$. Then we can build a set of quantized prompts in $G_{\delta,m_p} = \{0, \delta, 2\delta, ..., 1 - \delta\}^{d \times m_p}$ to index these $C$ functions. We denote this set of prompts as $\{\mathbf{P}_1, \mathbf{P}_2, ..., \mathbf{P}_C\}$. Finally we can create the quantized function $\bar{g}$ and let

$\bar{g}([\mathbf{P}_i, \mathbf{X}])_{:,m_p:} = \bar{f}_i(\mathbf{X})$ and $\bar{g}([\mathbf{P}_i, \mathbf{X}])_{:,:m_p} = 0, \forall \mathbf{X} \in [0,1]^{d \times m}, \mathbf{P} \in G_{\delta,m_p}$. For $\mathbf{P} \notin G_{\delta,m_p}$, we set $\bar{g}([\mathbf{P}, \mathbf{X}]) = \bar{g}([\mathbf{P}^*, \mathbf{X}])$ if $k_{i,j}\delta < \mathbf{P}_{i,j}, \mathbf{P}_{i,j}^* \leq (k_{i,j}+1)\delta$ and $\mathbf{P}^* \in G_{\delta,m_p}$.

Therefore, with a properly chosen $\delta = \delta_1$, for any $f \in \mathcal{F}_L$, we can find $\mathbf{P} \in \mathbb{R}^{d \times m_p}$ such that $d_P(f, \bar{g}([\mathbf{P}, \cdot])_{:,m_p:}) = d_p(\bar{f}, f) \leq \epsilon/2$.

$\square$

## C.2 Proof of Lemma 2

For any given quantized function $\bar{f} : [0,1]^{d \times m} \to [0,1]^{d \times m}$ with quantization at interval $\delta$, $\exists \bar{h} \in$

$\bar{\mathcal{T}}^{2,1,1}$ such that $\bar{f} = \bar{h}$ with positional embedding $\mathbf{E} = \begin{bmatrix} 0 & 1 & 2 & ... & m-1 \\ 0 & 1 & 2 & ... & m-1 \\ \vdots & \vdots & \vdots & \ddots & \vdots \\ 0 & 1 & 2 & ... & m-1 \end{bmatrix}$.

*Proof.* The proof is given following Section C in Yun et al. [2019a] appendix. With Section C.1 in Yun et al. [2019a], there exists a function $g_q$ composed of $\frac{dm}{\delta}$ token-wise feed-forward layers with hidden layer size $r = 1$ and ReLU activation to implement this scalar quantization on each input element:

$$g_q^{ent}(t) = \begin{cases} k\delta & \text{if } k\delta \leq t < (k+1)\delta, k = 0, 1, ..., m/\delta - 1 \\ -\delta^{-md} & \text{otherwise} \end{cases}$$

Then with Section C.2 in Yun et al. [2019a], we can stack $m(1/\delta)^d + 1$ attention layers to map all possible input sequences in grid $[0 : \delta : 1 - \delta]^d \times [1 : \delta : 2 - \delta]^d \times ... \times [m-1 : \delta : m - \delta]^d$ to distinct numbers which are at least $\delta$ from each other.

Finally we only require $O(m(1/\delta)^{dm})$ layers to map these distinct numbers to expected outputs. $\square$

## C.3 Proof of Lemma 3

Lemma 3 is alsmost the same as [Yun et al., 2019a] except that we use $\epsilon/2$ instead of $\epsilon/3$.

## C.4 Extension of Theorem 1 to Next-token Predictors

As an extension of Theorem 1, we consider approximating a set of sequence-to-sequence functions when we use a transformer layer as a next-token predictor. We consider a set of sequence-to-sequence functions $F_L$ with Lipschitz constant $L$ under norm $p$. $f \in F_L : [0,1]^{d \times m_1} \to [0,1]^{d \times m_2}$ accepts an input of length $m_1$ and outputs a sequence of length $m_2$. For any $\mathbf{X}, \mathbf{X}' \in [0,1]^{d \times m_1}$, we have $\|f(\mathbf{X}) - f(\mathbf{X}')\|_p \leq L\|\mathbf{X} - \mathbf{X}'\|_p$.

Next we show that we can construct a transformer $\tau$ which can approximate any $f \in F_L$ with prompt-tuning when we use it as a next-token predictor.

**Theorem 5.** *For any $f \in F_L$, we can construct a transformer $\tau$ such that for any $f \in F_L$, $1 \leq p < \infty$ and $\epsilon > 0$, we can find a prompt $\mathbf{P} \in [0,1]^{d \times m_p}$, such that $d_p(f, h(\mathbf{P})) \leq \epsilon$, where*

$$h(\mathbf{P}) = \tau_1([\mathbf{P}, \cdot])_{:,-1} \times \tau_2([\mathbf{P}, \cdot])_{:,-1} \times ... \times \tau_{m_2}([\mathbf{P}, \cdot])_{:,-1}.$$

*$\tau_i$ is the sequence-to-sequence function implemented with the transformer $\tau$ when accepting sequences with length $m_p + m_1 + i$.*

*Proof.* Similar to Theorem 1, we quantize the inputs to grid of $[0 : \delta : 1 - \delta]$ with interval $\delta$ and set $m_p = (\delta^{d \times m_2})^{\delta^{d \times m_1}}$. $\delta$ is chosen such that $L(m_1 d)^{1/p}\delta \leq m_2^{-1/p}\epsilon$. We index the $C = (\delta^{d \times m_2})^{\delta^{d \times m_1}}$ different $f$s as $f^1, f^2, ..., f^C$ and its sub-function to generate the $i$-th output token as $f_i^j$. The $C$ sequence-to-sequence functions can then be indexed by $C$ distinct prompts. Similar to Lemma 2, we can construct a transformer which can map all possible input sequences in grids $[0 : \delta : 1 - \delta] \times ... \times [m-1 : \delta : m - \delta]^d, 0 < m \leq m_1 + m_2 + m_p - 1$ to distinct numbers. A final series of MLP layers then map these distinct numbers to desired output vectors where inputs

in the same grid are mapped to the same output token at each step. Then for any input $\mathbf{x} \in [0,1]^{d \times m_1}$ and any $f^j$, we can find a prompt $\mathbf{P}_j$ such that

$$\|\tau([\mathbf{P}_j, \mathbf{x}])_{:,-1} - f_0^j(\mathbf{x})\|_p \le m_2^{-1/p}\epsilon$$
$$\|\tau([\mathbf{P}_j, \mathbf{x}, \tau([\mathbf{P}, \mathbf{x}])_{:,-1}])_{:,-1}, f_1^j([\mathbf{x}, \tau([\mathbf{P}, \mathbf{x}])_{:,-1}])\|_p \le m_2^{-1/p}\epsilon$$
$$...$$

Then we have $d_p(h(\mathbf{P}_j), f^j) \le \epsilon$. $\qquad\square$

## C.5    Proof of Lemma 4

If $\|\mathbf{W}_1\|_2 \times \|\mathbf{W}_2\|_2 < 1$, where $\|\cdot\|_2$ is the matrix spectral norm, then the `MLP` block in Definition 2 is invertible, ie, $\texttt{MLP}^{-1}$ is a singleton set.

*Proof.* Based on the sufficient conditions for invertibility of a residual block Behrmann et al. [2019], we have that if the feedforward part of a residual block $\mathbf{W}_2\texttt{ReLU}(\mathbf{W}_1\mathbf{X}_{:,1} + \mathbf{b}_1) + \mathbf{b}_2$ is a contraction with respect to some metric, i.e. its Lipschitz constant $< 1$, and the metric space on which is defined is complete, then `MLP` in eq 2 is invertible. Since we are dealing with the euclidean space, any metric induced by the $\|\cdot\|_p$ norm for $p \in [1, \infty]$ ensures the space is complete.
The Lipschitz constant of $\mathbf{W}_2\texttt{ReLU}(\mathbf{W}_1\mathbf{x} + \mathbf{b}_1) + \mathbf{b}_2$ is simply $\|\mathbf{W}_1\|_2 \times \|\mathbf{W}_2\|_2$. Thus the statement of the lemma follows. $\qquad\square$

## C.6    Proof of Theorem 2

For a single layer transformer $\tau$ defined above with Assumptions 1 and 2, we can build a seq-to-seq dataset $\{(\mathbf{X}_1 = [\mathbf{x}_1, \mathbf{x}_0], \mathbf{Y}_1 = [\mathbf{y}_{11}, \mathbf{y}_{10}]), (\mathbf{X}_2 = [\mathbf{x}_2, \mathbf{x}_0], \mathbf{Y}_2 = [\mathbf{y}_{21}, \mathbf{y}_{20}]))\}$, and we cannot find a prompt $\mathbf{P} \in \mathbb{R}^{d \times m_p}$ with any $m_p > 0$ such that $\tau([\mathbf{P}, \mathbf{X}_i]) = \mathbf{Y}_i$ holds for any $i = 1, 2$. The vectors $\mathbf{x}_0, \mathbf{x}_1, \mathbf{x}_2$ are denoted post positional encodings.

*Proof.* Before proving Theorem 2, we first provide a lemma that will be used in proof and also Theorem 3.

**Lemma 7.** *Given any* $\mathbf{c} \in \mathbb{R}^{d \times m}$, *there are* $\mathbf{x}_0$ *almost anywhere for which we can find another vector* $\mathbf{x}_1 \in \mathbb{R}^{d \times m}$ *such that* $\texttt{Att}(\mathbf{x}_0, [\mathbf{x}_0, \mathbf{x}_1]) \parallel \mathbf{c}$ *with full rank attention weights* $\mathbf{W}_q, \mathbf{W}_k, \mathbf{W}_v$.

*Proof.* If $\mathbf{W}_v\mathbf{x}_0 \parallel \mathbf{c}$, we can just set $\mathbf{x}_1 = \mathbf{x}_0$, which makes $\texttt{Att}(\mathbf{x}_0, [\mathbf{x}_0, \mathbf{x}_1]) \parallel \mathbf{c}$ hold.

If $\mathbf{W}_v\mathbf{x}_0 \nparallel \mathbf{c}$, let $\mathbf{v} = \alpha\mathbf{c} - \mathbf{W}_v\mathbf{x}_0$ where $\alpha \in \mathbb{R}$. As $\mathbf{W}_v$ is full-rank, we can find $\mathbf{x}$ such that $\mathbf{x} = \mathbf{W}_v^{-1}\mathbf{v} = \alpha\mathbf{W}_v^{-1}\mathbf{c} - \mathbf{x}_0$. Then we will have

$$\texttt{Att}(\mathbf{x}_0, [\mathbf{x}_0, \mathbf{x}_1])$$
$$= \frac{\exp\left((\mathbf{W}_q\mathbf{x}_0)^\top(\mathbf{W}_k\mathbf{x}_0)\right)\mathbf{W}_v\mathbf{x}_0 + \exp\left((\mathbf{W}_q\mathbf{x}_0)^\top(\mathbf{W}_k(\alpha\mathbf{W}_v^{-1}\mathbf{c} - \mathbf{x}_0))\right)(\alpha\mathbf{c} - \mathbf{W}_v\mathbf{x}_0)}{\exp\left((\mathbf{W}_q\mathbf{x}_0)^\top(\mathbf{W}_k\mathbf{x}_0)\right) + \exp\left((\mathbf{W}_q\mathbf{x}_0)^\top(\mathbf{W}_k(\alpha\mathbf{W}_v^{-1}\mathbf{c} - \mathbf{x}_0))\right)}$$

Therefore, as long as $\mathbf{W}_q\mathbf{x}_0 \not\perp \mathbf{W}_k(\mathbf{W}_v^{-1}\mathbf{c})$, we can change $\alpha$ such that $\texttt{Att}(\mathbf{x}_0, [\mathbf{x}_0, \mathbf{x}_1]) = \beta\mathbf{W}_v\mathbf{x}_0 + (1-\beta)(\alpha\mathbf{c} - \mathbf{W}_v\mathbf{x}_0)$ where $\beta = \frac{\exp\left((\mathbf{W}_q\mathbf{x}_0)^\top(\mathbf{W}_k\mathbf{x}_0)\right)}{\exp\left((\mathbf{W}_q\mathbf{x}_0)^\top(\mathbf{W}_k\mathbf{x}_0)\right) + \exp\left((\mathbf{W}_q\mathbf{x}_0)^\top(\mathbf{W}_k(\alpha\mathbf{W}_v^{-1}\mathbf{c} - \mathbf{x}_0))\right)}$. When $\alpha = 0$, $\texttt{Att}(\mathbf{x}_0, [\mathbf{x}_0, \mathbf{x}_1]) = \mathbf{W}_v\mathbf{x}_0$, when $\alpha \to -\infty$ or $\alpha \to \infty$, $\texttt{Att}(\mathbf{x}_0, [\mathbf{x}_0, \mathbf{x}_1]) \to \alpha\mathbf{c} - \mathbf{W}_v\mathbf{x}_0$. As $\texttt{Att}(\mathbf{x}_0, [\mathbf{x}_0, \mathbf{x}_1])$ is continuous w.r.t changing $\alpha$, there must exist an $\alpha$ such that $\texttt{Att}(\mathbf{x}_0, [\mathbf{x}_0, \mathbf{x}_1]) \parallel \mathbf{c}$. $\qquad\square$

Pass the two input sequences $\mathbf{X}_1, \mathbf{X}_2$ through the attention layer $\texttt{Att}$ with any prompt $\mathbf{P}$, we can get the last output token as:

$$\texttt{Att}(\mathbf{x}_0, [\mathbf{P}, \mathbf{X}_1]) = \lambda(\mathbf{X}_1, \mathbf{x}_0, [\mathbf{P}, \mathbf{X}_1])\texttt{Att}(\mathbf{x}_0, \mathbf{X}_1) + \lambda(\mathbf{P}, \mathbf{x}_0, [\mathbf{P}, \mathbf{X}_1])\texttt{Att}(\mathbf{x}_0, \mathbf{P}) \qquad (12)$$
$$\texttt{Att}(\mathbf{x}_0, [\mathbf{P}, \mathbf{X}_2]) = \lambda(\mathbf{X}_2, \mathbf{x}_0, [\mathbf{P}, \mathbf{X}_2])\texttt{Att}(\mathbf{x}_0, \mathbf{X}_2) + \lambda(\mathbf{P}, \mathbf{x}_0, [\mathbf{P}, \mathbf{X}_2])\texttt{Att}(\mathbf{x}_0, \mathbf{P}) \qquad (13)$$

Here $\lambda(\mathbf{X}_1, \mathbf{x}_2, \mathbf{X}_3 = [\mathbf{X}_1, \mathbf{X}_2]) \in (0, 1)$ is a positive scalar, defined as

$$\lambda(\mathbf{X}_1, \mathbf{x}_2, \mathbf{X}_3) = \frac{\sum_j \exp((\mathbf{W}_k \mathbf{x}_{1j})^\top (\mathbf{W}_q \mathbf{x}_2))}{\sum_j \exp((\mathbf{W}_k \mathbf{x}_{3j})^\top (\mathbf{W}_q \mathbf{x}_2))}.$$

$\mathbf{x}_{ij}$ is the $j$th token in $\mathbf{X}_i$ for notation simplicity.

1. Then from equation 12, $\mathtt{Att}(\mathbf{x}_0, \mathbf{P})$ must be on $\mathtt{Cone}(-\mathtt{Att}(\mathbf{x}_0, \mathbf{X}_1), \mathtt{Att}(\mathbf{x}_0, [\mathbf{P}, \mathbf{X}_1]))$ and $\mathtt{Cone}(-\mathtt{Att}(\mathbf{x}_0, \mathbf{X}_2), \mathtt{Att}(\mathbf{x}_0, [\mathbf{P}, \mathbf{X}_2]))$.

2. On the otherhand, as we want to memorize the two examples, we must have $\mathtt{Att}(\mathbf{x}_0, [\mathbf{P}, \mathbf{X}_1]) + \mathbf{x}_0 \in \mathtt{MLP}^{-1}(\mathbf{y}_{10})$ and $\mathtt{Att}(\mathbf{x}_0, [\mathbf{P}, \mathbf{X}_2]) + \mathbf{x}_0 \in \mathtt{MLP}^{-1}(\mathbf{y}_{20})$.

We construct the dataset $S$ with arbitrary $\mathbf{x}_0, \mathbf{y}_{10}$ and $\mathbf{y}_{20}$. Then if $\dim((\mathtt{MLP}^{-1}(\mathbf{y}_{10}) - \mathbf{x}_0) \cup (\mathtt{MLP}^{-1}(\mathbf{y}_{20}) - \mathbf{x}_0)) + 2 \le d$ (Assumption 2), we can find two vectors $\mathbf{c}_1, \mathbf{c}_2$ such that $\mathbf{c}_1, \mathbf{c}_2 \perp \mathbf{v}$ : $\mathbf{v} + \mathbf{x}_0 \in \mathtt{MLP}^{-1}(\mathbf{y}_{10})$ or $\mathbf{v} + \mathbf{x}_0 \in \mathtt{MLP}^{-1}(\mathbf{y}_{20})$ and $\mathbf{c}_1 \perp \mathbf{c}_2$. Then we can choose $\mathbf{x}_1, \mathbf{x}_2$ such that $\mathtt{Att}(\mathbf{x}_0, \mathbf{X}_1) \parallel \mathbf{c}_1$ and $\mathtt{Att}(\mathbf{x}_0, \mathbf{X}_2) \parallel \mathbf{c}_2$ (Lemma 7). Combine this construction with assumption 1, we have that $\mathtt{Cone}(-\mathtt{Att}(\mathbf{x}_0, \mathbf{X}_1), \mathtt{Att}(\mathbf{x}_0, [\mathbf{P}, \mathbf{X}_1]))$ and $\mathtt{Cone}(-\mathtt{Att}(\mathbf{x}_0, \mathbf{X}_2), \mathtt{Att}(\mathbf{x}_0, [\mathbf{P}, \mathbf{X}_2]))$ has no intersection, which means that we cannot find a $\mathbf{P}$ to memorize this constructed dataset. $\square$

### C.7 Proof of Lemma 5

For a standard single-layer transformer $\tau$ defined in Definition 2 with $r \ge n$ MLP hidden neurons, for any sequence-to-sequence dataset $S$ satisfying Assumptions 1, we can apply a low-rank update to MLP weights with $O(nd)$ parameters to memorize $\tau(\mathbf{X}_i)_{:,m} = \mathbf{y}_{im}$.

*Proof.* We use $\mathtt{MLP}_j(\mathbf{x})$ to denote the $jth$ output of the MLP layer for an input token $\mathbf{x}$, which is

$$\mathtt{MLP}_j(\mathbf{x}) = x_j + b_{2,j} + \sum_{k=1}^{m} w_{k,j} \max(\langle \mathbf{a}_k, \mathbf{x} \rangle + b_{1,k}, 0)$$

According to our assumption, $\mathtt{Att}(\mathbf{x}_{im}, \mathbf{X}_i)$ are unique vectors for $i = 1, 2, ..., n$. Then we only need to use the MLP layer to map each $\mathbf{x}_i = \mathtt{Att}(\mathbf{x}_{im}, \mathbf{X}_i) + \mathbf{x}_{im}$ to $\mathbf{y}_{im}$, where we get a new token-wise dataset $\{(\mathbf{x}_1, \mathbf{y}_1), (\mathbf{x}_2, \mathbf{y}_2), ..., (\mathbf{x}_n, \mathbf{y}_n)\}$

Then we need to find $w_k, \mathbf{a}_k$ and $b_k$ such that

$$\mathtt{MLP}_j(\mathbf{x}_i) = x_{i,j} + b_{2,j} + \sum_{k=1}^{m} w_{k,j} \max(\langle \mathbf{a}_k, \mathbf{x}_i \rangle + b_{1,k}, 0) = y_{i,j}, i = 1, 2, ..., n, j = 1, 2, ..., d \tag{14}$$

, which is equivalent to constructing a standard MLP to memorize a dataset:

$$\sum_{k=1}^{n} w_{k,j} \max(\langle \mathbf{a}_k, \mathbf{x}_i \rangle + b_{1,k}, 0) = y_{i,j} - x_{i,j} - \sum_{k=n+1}^{m} w_{k,j} \max(\langle \mathbf{a}_k, \mathbf{x}_i \rangle + b_{1,k}, 0) - b_{2,j} \tag{15}$$

Follow Thoerem 1 in Yun et al. [2019b], we can construct $\mathbf{a}, b_1, ..., b_n$ such that for $\mathbf{x}_1, \mathbf{x}_2, ..., \mathbf{x}_n$, we have $z_i = \langle \mathbf{a}, \mathbf{x}_i \rangle$, $b_1 < z_1 < b_2 < ... < b_n < z_n$. Then we can find $w_1, ..., w_n$ which solves equation 15. For $d$-dimension output, we need to find $\mathbf{W} \in \mathbb{R}^{n \times d}$ and $\mathbf{a} \in \mathbb{R}^d$ and $\mathbf{b} \in \mathbb{R}^n$. With LoRA, we need a low-rank update of size $m \times n + n \times d$ for $\mathbf{W}_2$, a low-rank update of size $d \times n + n \times m$ for $\mathbf{W}_1$ and an update of size $n$ for $\mathbf{b}_1$, which is $O(n \times d)$ in total. Normally we have $m \simeq d$, then we need an update with parameter size around $(4n + 1)d$ to memorize the last token of $n$ training examples. $\square$

### C.8 Proof of Theorem 3

For any single layer transformer $\tau$ defined in Definition 2, there exists a seq-to-seq dataset $\{(\mathbf{X}_1 = [\mathbf{x}_{10}, \mathbf{x}_1], [\mathbf{y}_{10}, \mathbf{y}_{11}]), (\mathbf{X}_2 = [\mathbf{x}_{20}, \mathbf{x}_2], [\mathbf{y}_{20}, \mathbf{y}_{21}]), ..., (\mathbf{X}_n = [\mathbf{x}_{n0}, \mathbf{x}_n], [\mathbf{y}_{n0}, \mathbf{y}_{n1}])\}$ that satisfies Assumption 1 with $n < d$ training examples such that we need at least $n$ prompt tokens in $\mathbf{P}$ to memorize the training set, ie, for $\tau([\mathbf{P}, \mathbf{X}_i])_{:,-1} = \mathbf{y}_{i1}$ to hold for all $i = 1, 2, ..., n$.

*Proof.* Without loss of generality, we assume $\mathbf{W}_2$ has no zero elements, otherwise we can just ignore this hidden neuron in MLP layer.

$\mathbb{R}^d$ has $d$ bases $\{\mathbf{t}_j : j = 1, 2, ..., d\}$, then $\texttt{MLP}^{-1}(\mathbf{y}_{i1})$ must be bounded on either positive or negative part of these $d$ directions, which means there exists $B \geq 0$ such that

$$\frac{\mathbf{v}^\top \mathbf{t}_j}{\|\mathbf{t}_j\|} \leq B, \forall \mathbf{v} \in \texttt{MLP}^{-1}(\mathbf{y}_{i1}), j = 1, 2, ..., d$$

Otherwise $\forall B > 0$, $\exists \mathbf{t}_j$, we can find a $\mathbf{v} \in \texttt{MLP}^{-1}(\mathbf{y}_i)$ that $\frac{\mathbf{v}^\top \mathbf{t}_j}{\|\mathbf{t}_j\|} \geq B$. Meanwhile we have $\texttt{MLP}(\mathbf{v}) = \mathbf{v} + \mathbf{b}_2 + \mathbf{W}_2\texttt{ReLU}(\mathbf{W}_1\mathbf{v} + \mathbf{b}_1)$. As $\|\mathbf{v}\|$ can be arbitrarily large, if $\mathbf{W}_1\mathbf{v} = \mathbf{0}$, $\|\texttt{MLP}(\mathbf{v})\| \to \infty$ if $\|\mathbf{v}\| \to \infty$. if $\mathbf{W}_1\mathbf{v} \neq \mathbf{0}$, $\|\texttt{MLP}(\mathbf{v})\|$ can also be arbitrarily large when increasing the norm of $\mathbf{v}$ due to the non-linearity of $\texttt{ReLU}(\mathbf{W}_1\mathbf{v} + \mathbf{b}_1)$.

Then we can find a set of $n$ linearly independent vectors $\{\mathbf{c}_1, \mathbf{c}_2, ..., \mathbf{c}_n\}$ such that $\{\mathbf{a}_i : \mathbf{a}_i - \mathbf{c}_i \perp \mathbf{c}_i, \mathbf{a}_i \in \texttt{MLP}^{-1}(\mathbf{y}_{i1})\} = \emptyset$ by enlarging the norm of $\mathbf{c}_i$. With the $n$ $\mathbf{c}_i$ vectors, we can begin to construct our dataset:

We set $\mathbf{x}_i = \mathbf{c}_i, i = 1, 2, ..., n$ and find $\mathbf{x}_{i0}$ such that $\mathbf{c}_i \perp \texttt{Att}(\mathbf{x}_i, \mathbf{X}_i)$ (Lemma 7) and $\texttt{Att}(\mathbf{x}_i, \mathbf{X}_i)$ are distinct for $i = 1, 2, ..., n$ (Assumption 1), which makes $\{\mathbf{a}_1 - \mathbf{x}_1 - \lambda(\mathbf{X}_1, \mathbf{x}_1, [\mathbf{P}, \mathbf{X}_1])\texttt{Att}(\mathbf{x}_1, \mathbf{X}_1), ..., \mathbf{a}_n - \mathbf{x}_n - \lambda(\mathbf{X}_n, \mathbf{x}_n, [\mathbf{P}, \mathbf{X}_n])\texttt{Att}(\mathbf{x}_n, \mathbf{X}_n)\}$ linearly independent for any $\mathbf{a}_i \in \texttt{MLP}^{-1}(\mathbf{y}_{i1})$. Here $\lambda(\cdot, \cdot, \cdot)$ is the same as defined in Section C.6.

Moreover, we have

$$\texttt{Att}(\mathbf{x}_i, [\mathbf{P}, \mathbf{X}_i]) = \lambda(\mathbf{X}_i, \mathbf{P}, [\mathbf{P}, \mathbf{X}_i])\texttt{Att}(\mathbf{x}_i, \mathbf{P}) + \lambda(\mathbf{X}_i, \mathbf{x}_i, [\mathbf{P}, \mathbf{X}_i])\texttt{Att}(\mathbf{x}_i, \mathbf{X}_i) \quad (16)$$
$$\in \texttt{MLP}^{-1}(\mathbf{y}_{i1}) - \mathbf{x}_i$$

Then $\texttt{Att}(\mathbf{x}_i, \mathbf{P}), i = 1, 2, ..., n$ must be $n$ linearly independent vectors, which requires

$$\texttt{rank}(\mathbf{W}_v\mathbf{PA}) = n, \quad (17)$$

where $\mathbf{A} \in \mathbb{R}^{m_p \times n}$ is the attention score matrix between $\mathbf{x}_i$ and $\mathbf{P}$. $\mathbf{P} \in \mathbb{R}^{d \times m_p}$ is the prompt token sequence and $\mathbf{W}_v$ is the attention value weight. Therefore, we must have $m_p \geq n$. $\quad \square$

## C.9 Proof of Theorem 4

A transformer $\mathcal{T}$ is invertible if:

1. The Lipschitz constant of the attention block in each layer $\tau^l$ is *strictly* less than 1
2. The Lipschitz constant of the 2-layer ReLU block in each layer $\tau^l$, which is bounded by $\|\mathbf{W}_2^l\|_2 \times \|\mathbf{W}_1^l\|_2$, is *strictly* less than 1

*Proof.* This proof is based on the proof provided for lemma 4, thus we restrict ourselves to the sketch: Based on the sufficient condition for invertibility in Behrmann et al. [2019], condition (1) implies that the attention block (eq 1) with the residual connection, ie $\mathbf{X} + \texttt{Att}(\mathbf{X}, \mathbf{X})$, is an invertible function. Similarly, condition (2) implies that the MLP block which constitutes of the 2-layer ReLU block with the residual connection (eq 2) also exhibit invertibility.
Thus each transformer layer $\tau^l$ (eq 3) is invertible by noting that its a composition of two invertible functions. The same property ensures that the entire transformer architecture $\mathcal{T}$ is also invertible. $\quad \square$

## C.10 Proof of Lemma 6

Under the compactness condition, the Lipschitz constant of the $i$-th attention head in the $l$-th transformer layer, denoted for simplicity as $\texttt{Att}^{i,l}$, admits the following bound w.r.t the entire input sequence of length $m$:

$$Lip(\texttt{Att}^{i,l}(\cdot, \cdot)) \leq (1 + 8\sqrt{m}(D^l)^2\|(\mathbf{W}_k^{i,l})^T\mathbf{W}_q^{i,l}\|_2)\|\mathbf{W}_v^{i,l}\|_2, \quad (18)$$

and the Lipschitz constant of the entire attention block in layer $l$, denoted as $\texttt{Att}^l$, admits the bound:

$$Lip(\texttt{Att}^l(\cdot, \cdot)) \leq \sqrt{\sum_{i=1}^{h}(\|\mathbf{W}_o^{i,l}\|_2 \times Lip(\texttt{Att}^{i,l}))^2}. \quad (19)$$

*Proof.* We drop the superscripts $i, l$ in the proof to avoid notation clutter. Similarly, we denote the concatenation of the prompt matrix $\mathbf{P}$ and the original input matrix $\mathbf{X}$, simply with $\mathbf{X}$.

**Derivation for single head eq 18:**
Consider two matrices $\mathbf{X}_1, \mathbf{X}_2 \in \mathcal{X} = \{\mathbf{X} \in \mathbb{R}^{d \times m}; \|\mathbf{X}\|_2 \leq D\}$. Denote with $\mathbf{A}_1, \mathbf{A}_2$ the corresponding attention matrices respectively, which can be defined as:

$$\mathbf{A}_1 = \sigma((\mathbf{W}_k \mathbf{X}_1)^\top \mathbf{W}_q \mathbf{X}_1)$$
$$\mathbf{A}_2 = \sigma((\mathbf{W}_k \mathbf{X}_2)^\top \mathbf{W}_q \mathbf{X}_2) \tag{20}$$

The output of the attention head, denoted with $Att(\cdot)$ admits the following:

$$\|Att(\mathbf{X}_1) - Att(\mathbf{X}_2)\|_2 = \|\mathbf{W}_v \mathbf{X}_1 \mathbf{A}_1 - \mathbf{W}_v \mathbf{X}_2 \mathbf{A}_2\|_2 \tag{21}$$

$$\overset{a}{\leq} \|\mathbf{X}_1 \mathbf{A}_1 - \mathbf{X}_2 \mathbf{A}_2\|_2 \|\mathbf{W}_v\|_2 \tag{22}$$

$$= \|\mathbf{X}_1 \mathbf{A}_1 - \mathbf{X}_2 \mathbf{A}_1 + \mathbf{X}_2 \mathbf{A}_1 - \mathbf{X}_2 \mathbf{A}_2\|_2 \|\mathbf{W}_v\|_2 \tag{23}$$

$$\leq (\|\mathbf{A}_1\|_2 \|\mathbf{X}_1 - \mathbf{X}_2\|_2 + \|\mathbf{X}_2\|_2 \|\mathbf{A}_1 - \mathbf{A}_2\|_2) \|\mathbf{W}_v\|_2 \tag{24}$$

$$\overset{b}{\leq} (\|\mathbf{X}_1 - \mathbf{X}_2\|_2 + \|\mathbf{A}_1 - \mathbf{A}_2\|_2 D) \|\mathbf{W}_v\|_2 \tag{25}$$

where $(a)$ holds from the spectral norm properties and in $(b)$ we use the bounded input spectral norm assumptions.
We now focus on the second term $\|\mathbf{A}_1 - \mathbf{A}_2\|_2$ in eq 25. From the bound in lemma 9, we have:

$$\|\mathbf{A}_1 - \mathbf{A}_2\|_2 \leq 2\sqrt{m} \|\mathbf{G}\|_2 \tag{26}$$

where $\mathbf{G}$ is the diagonal matrix with entires described in lemma 8
We can now invoke lemma 10 to obtain the following :

$$\|\mathbf{A}_1 - \mathbf{A}_2\|_2 \leq 2\sqrt{m} \times 2 \times 2\|\mathbf{W}_k^T \mathbf{W}_q\|_2 D \times \|\mathbf{X}_1 - \mathbf{X}_2\|_2 \tag{27}$$

Combining the previous inequality with eq 25, we have the following bound:

$$\|Att(\mathbf{X}_1) - Att(\mathbf{X}_2)\|_2 \leq (1 + 8\sqrt{m} \|\mathbf{W}_k^T \mathbf{W}_q\|_2 D^2) \|\mathbf{W}_v\|_2 \|\mathbf{X}_1 - \mathbf{X}_2\|_2 \tag{28}$$

**Derivation for the entire block eq 11:**
The proof follows simply by leveraging the following property:
*Property:* for a matrix $\mathbf{C} = [\mathbf{A}, \mathbf{B}]$, the spectral norm of $\mathbf{C}$ admits the bound:

$$\|\mathbf{C}\|_2 \leq \sqrt{\|\mathbf{A}\|_2^2 + \|\mathbf{B}\|_2^2}$$

We then simply combine the definition of the attention block and the lipschitz constant bound in eq 18 with the above property in order to obtain the desired bound. $\qquad \square$

**Lemma 8** (Dong et al. [2021] Lemma A.1). *For the column stochastic matrix $\mathbf{A}_1$ obtained by performing column-wise softmax of some matrix $\mathbf{Z}_1$ (where in our setting $\mathbf{Z}_1 = (\mathbf{W}_k \mathbf{X}_1)^\top \mathbf{W}_q \mathbf{X}_1$, and another row stochastic matrix $\mathbf{A}_2$ obtained by performing column-wise softmax of some matrix $\mathbf{Z}_2$, where $\mathbf{Z}_2 = \mathbf{Z}_1 - \mathbf{E}$ (for some $\mathbf{E}$, which **need not** belong to $\mathcal{X}$), we have the following bound:*

$$\mathbf{A}_2(\mathbf{I} - \mathbf{G}) \leq \mathbf{A}_1 \leq \mathbf{A}_2(\mathbf{I} + 2\mathbf{G}) \tag{29}$$

*where the inequality is elementwise and $\mathbf{G}$ is a diagonal matrix with entries as $\mathbf{G}_{ii} = \max_{j,j'} |\delta_i^T \mathbf{E}(\delta_j^T - \delta_{j'}^T)|$. Here $\delta_k$ is a one-hot vector with the entry 1 in the $k^{th}$ dimension.*

**Lemma 9.** *Following the notations of lemma 8, we have the following spectral norm bound:*

$$\|\mathbf{A}_1 - \mathbf{A}_2\|_2 \leq 2\sqrt{m} \|\mathbf{G}\|_2 \tag{30}$$

*Proof.* We begin by noting the following entry-wise inequality from eq 29:

$$\mathbf{A}_2 \mathbf{G} \leq \mathbf{A}_1 - \mathbf{A}_2 \leq 2\mathbf{A}_2 \mathbf{G} \tag{31}$$

which ensures that $\|\mathbf{A}_1 - \mathbf{A}_2\|_F \leq 2\|\mathbf{A}_2\mathbf{G}\|_F$.
We also have the following using matrix norm equivalence:

$$\|\mathbf{A}_1 - \mathbf{A}_2\|_2 \leq \|\mathbf{A}_1 - \mathbf{A}_2\|_F \tag{32}$$

Invoking the matrix norm equivalence again, we have that

$$2\|\mathbf{A}_2\mathbf{G}\|_F \leq 2\sqrt{rank(\mathbf{A}_2\mathbf{G})}\|\mathbf{A}_2\mathbf{G}\|_2 \tag{33}$$

where $rank(\cdot)$ is the matrix rank.
Combining the inequalities, we attain the bound :

$$\|\mathbf{A}_1 - \mathbf{A}_2\|_2 \leq 2\sqrt{m}\|\mathbf{A}_2\|_2\|\mathbf{G}\|_2 \tag{34}$$

since $\mathbf{A}_2$ is column-stochastic , $\|\mathbf{A}_2\|_2 = 1$ $\qquad\square$

**Lemma 10.** *The term* $\mathbf{G}$ *in lemma 9 admits the following spectral norm bound:*

$$\|\mathbf{G}\|_2 \leq 2D\|\mathbf{W}_q\mathbf{W}_k^T\|_2\|\mathbf{X}_1 - \mathbf{X}_2\|_2 \tag{35}$$

*here* $D$ *is the previously stated spectral norm bound of the inputs* $\mathbf{X}^l \in \mathcal{X}^l$.

*Proof.* We begin by noting that since $\mathbf{G}$ is a square diagonal matrix with non-negative real values, the singular values of $\mathbf{G}$ are the corresponding diagonal elements.
We thus have that $\|\mathbf{G}\|_{max} = \|\mathbf{G}\|_2$ , where $\|\cdot\|_{max}$ is the $max$ norm.
Since $\mathbf{G}$ admits the form described in lemma 8, it is trivial to note that:

$$\|\mathbf{G}\|_{max} = \max_{i,j,i',j'} |\mathbf{E}_{i,j} - \mathbf{E}_{i',j'}| \tag{36}$$

$$\leq 2\|\mathbf{E}\|_{max} \leq 2\|\mathbf{E}\|_2 \tag{37}$$

where the second inequality follows from the matrix norm equivalence.
Now, we can bound the last term $\|\mathbf{E}\|_2$ by noting that the inputs $\mathbf{X}$ belong to a bounded set. This allows us to provide the following bounds:

$$\|\mathbf{E}\|_2 = \|(\mathbf{W}_k\mathbf{X}_1)^\top\mathbf{W}_q\mathbf{X}_1 - (\mathbf{W}_k\mathbf{X}_2)^\top\mathbf{W}_q\mathbf{X}_2\|_2 \tag{38}$$

$$= \|(\mathbf{W}_k\mathbf{X}_1)^\top\mathbf{W}_q\mathbf{X}_1 - (\mathbf{W}_k\mathbf{X}_1)^\top\mathbf{W}_q\mathbf{X}_2 + (\mathbf{W}_k\mathbf{X}_1)^\top\mathbf{W}_q\mathbf{X}_2 - (\mathbf{W}_k\mathbf{X}_2)^\top\mathbf{W}_q\mathbf{X}_2\|_2 \tag{39}$$

$$\leq (\|\mathbf{X}_1\|_2\|\mathbf{W}_k^T\mathbf{W}_q\|_2 + \|\mathbf{X}_2\|_2\|\mathbf{W}_k^T\mathbf{W}_q\|_2)\|\mathbf{X}_1 - \mathbf{X}_2\|_2 \tag{40}$$

$$\leq 2D\|\mathbf{W}_k^T\mathbf{W}_q\|_2\|\mathbf{X}_1 - \mathbf{X}_2\|_2 \tag{41}$$

$$\square$$

### C.11 Extension of Lemma 6

Lemma 6 and theorem 4 operate over functions from $\mathcal{P}^1 \times \mathcal{X}^1 \to \mathcal{P}^{L+1} \times \mathcal{X}^{L+1}$. We can relax the requirement of the prompt and provide the Lipschitz constant upper bound in consideration to functions of the form $\mathcal{X}^1 \to \mathcal{X}^{L+1}$ by using the following assumption:

**Assumption 3.** *Assume for simplicity that* $\|\mathbf{P}_1^l - \mathbf{P}_2^l\|_2 \leq \alpha^l\|\mathbf{X}_1^l - \mathbf{X}_2^l\|_2; \forall l \geq 1$. $\alpha^l = 0$ *when* $l = 1$.

**Note:** A recursive expression for $\alpha^l$ in the above assumption can be provided, but the expression does not admit a simplified form and we thus omit it here.

We will use $\mathcal{D}_X^l$ , akin to eq 9, to denote the compactness corresponding to the input matrix across the layers.
Based on this assumption, we have the following Lipschitz constant upper bound:

**Lemma 11.** *The Lipschitz constant of the single head* $\mathit{Att}^{i,l}$ *admits the following bound w.r.t the input part,* $\mathbf{X}^1$ *of length* $m_X$, *of the input sequence:*

$$Lip(\mathit{Att}^{i,l}(\cdot,\cdot)) \leq \left(\sqrt{1 + (\alpha^l)^2} + 8\sqrt{m_X}(D_X^l)^2(1 + (\alpha^l)^2)\|(\mathbf{W}_k^{i,l})^T\mathbf{W}_q^{i,l}\|_2\right)\|\mathbf{W}_v^{i,l}\|_2 \tag{42}$$

*For* $l = 1$, $\alpha^l = 0$ *in the above bound.*
*The Lipschitz constant of the entire attention block in layer* $l$ *follows similarly.*

*Proof.* For some first layer input $\mathbf{X}_1^1$ and prompt $\mathbf{P}$, let us denote the direct output of the attention head in the $l$-th layer with $\overrightarrow{\mathbf{X}}_1^l$. We have the following update rule for $\overrightarrow{\mathbf{X}}_1^l$:

$$\overrightarrow{\mathbf{X}}_1^l = \mathbf{W}_v^{i,l}[\mathbf{P}_1^l, \mathbf{X}_1^l] \cdot \sigma((\mathbf{W}_k^{i,l}[\mathbf{P}_1^l, \mathbf{X}_1^l])^\top \mathbf{W}_q^i \mathbf{X}_1^l) = \mathbf{W}_v^{i,l}[\mathbf{P}_1^l, \mathbf{X}_1^l] \cdot \mathbf{A}_1^l \tag{43}$$

Here, $\mathbf{P}_1^l$ is the updated prompt matrix w.r.t the input. For two different inputs $\mathbf{X}_1^1$ and $\mathbf{X}_2^1$ at the first layer, $\mathbf{P}_1^1 = \mathbf{P}_2^1 = \mathbf{P}$, since the prompt is same across all inputs. $\mathbf{A}_1^l$ is then simply the corresponding column-stochastic matrix.

With the context clear, we now drop the superscripts $i, l$, as done previously. For $\|\overrightarrow{\mathbf{X}}_1 - \overrightarrow{\mathbf{X}}_2\|_2$, we have:

$$\|\overrightarrow{\mathbf{X}}_1 - \overrightarrow{\mathbf{X}}_2\|_2 \leq \|[\mathbf{P}_1, \mathbf{X}_1]\mathbf{A}_1 - [\mathbf{P}_2, \mathbf{X}_2]\mathbf{A}_2\|_2 \|\mathbf{W}_v\|_2$$

$$\leq \left( \|\mathbf{A}_1\|_2 \sqrt{\|\mathbf{P}_1 - \mathbf{P}_2\|_2^2 + \|\mathbf{X}_1 - \mathbf{X}_2\|_2^2} + \sqrt{\|\mathbf{P}_2\|_2^2 + \|\mathbf{X}_2\|_2^2} \|\mathbf{A}_1 - \mathbf{A}_2\|_2 \right) \|\mathbf{W}_v\|_2 \tag{44}$$

where the second inequality is attained using the property of spectral norm of concatenated matrices.

We now consider the term $\|\mathbf{A}_1 - \mathbf{A}_2\|_2$. By invoking lemmas 9 and 10, we have that:

$$\|\mathbf{A}_1 - \mathbf{A}_2\|_2 \leq 2\sqrt{m_X} \times 2\|\mathbf{E}\|_2$$
$$\text{where} \quad \mathbf{E} = (\mathbf{W}_k[\mathbf{P}_1, \mathbf{X}_1])^\top \mathbf{W}_q \mathbf{X}_1 - (\mathbf{W}_k[\mathbf{P}_2, \mathbf{X}_2])^\top \mathbf{W}_q \mathbf{X}_2 \tag{45}$$

Invoking assumption 3 , $\|\mathbf{E}\|_2$ can further be bounded as:

$$\|\mathbf{E}\|_2 \leq 2D_X \sqrt{1 + \alpha^2} \|(\mathbf{W}_k)^T \mathbf{W}_q\|_2 \tag{46}$$

Finally, by combining the bound for $\|\mathbf{E}\|_2$ and assumption 3 with eq 44, we obtain:

$$\|\overrightarrow{\mathbf{X}}_1 - \overrightarrow{\mathbf{X}}_2\|_2 \tag{47}$$
$$\leq \left( \sqrt{1 + \alpha^2} + D_X \sqrt{1 + \alpha^2} \times 8\sqrt{m_X} D_X \sqrt{1 + \alpha^2} \|(\mathbf{W}_k)^T \mathbf{W}_q\|_2 \right) \|\mathbf{W}_v\|_2 \|\mathbf{X}_1 - \mathbf{X}_2\|_2$$
$$\tag{48}$$

which provides us the desired bound. $\qquad\square$

By setting $\alpha^l = 0$, the case when there is no prompt, we obtain a similar bound as lemma 6

