# OpenReview forum: "Universality and Limitations of Prompt Tuning"
_NeurIPS.cc/2023/Conference — NeurIPS 2023 poster_

### Official Review · Reviewer_AVty · 2023-07-06

**Soundness:** 3 good
**Presentation:** 3 good
**Contribution:** 2 fair
**Rating:** 6
**Confidence:** 3

**Summary:**

This paper examines the theoretical capacity and limitations of prompt tuning. The authors have demonstrated the possibility of constructing a large transformer model that is sufficient for prompt-tuning to exhibit universal approximation over a Lipschitz function space. However, they have also shown the limitations of prompt tuning compared to model fine-tuning by constructing sequence-to-sequence datasets that cannot be learned by prompt-tuning with a given single-layer transformer, even if the soft-prompt embedding size approaches infinity. Additionally, compared with LoRA, prompt-tuning may require more prompt tokens for exact memorization.

**Strengths:**

- Originality
    - This paper has pretty good originality. It first theoretically studies the theoretical capacity and limitations of prompt tuning.

- Clarity
    - The structures are clear and conclusions are emphasized.

- Significance
    - The problem studied in this paper is important

**Weaknesses:**

- The authors have conducted experiments to support the proposed limitation. However, additional experimental verifications may be necessary to reinforce the theoretical claims and demonstrate the impact on a wider range of tasks and datasets.

- The authors have created sequence-to-sequence datasets that are simple in structure but cannot be memorized by prompt-tuning for a given transformer model. Therefore, they argue that prompt-tuning has limited expressiveness compared to model fine-tuning. However, is memorization typically necessary in real-world applications? Does memorization lead to overfitting? As I understand it, such a phenomenon could also be viewed as a potential advantage in preventing overfitting and achieving generalization ability in the process of downstream adaptation.

- The authors have provided quite a few mathematical proofs and concluded several insights. I wonder whether there are any guidance for practical prompt-tuning applications.  For example, they provide a lower bound on the required number of prompt tokens for exact memorization and point out that such lower bound can be higher than LoRA’s. I would like to know that on which scenarios this could happen and it is recommended to user LoRA rather than prompt-tuning.

- The authors need to correct some minor typos.
    - line 30, typo:  the or a
    - line 503, line 508, are they the same reference?

**Questions:**

Please respond to questions in the "Weaknesses" section.

**Limitations:**

some assumptions are given and not sure whether they are applied in real-world applications

---

> ### Author Rebuttal · Authors · 2023-08-10
>
> We sincerely thank Reviewer AVty for the feedback and valuable suggestions. We have addressed each of the concerns below:
>
> **Regarding additional experiments on more datasets**
>
> Please see our general response for experiments on additional datasets
>
> **Significance of memorization capacity of neural networks**
>
> Memorization capacity is a measurement of the representation power of a neural network architecture. For universal approximators like MLP networks, we can control their memorization capability (expressive power) by changing its depth or breadth to avoid underfitting on complicated data. However, as shown in our theorems, prompt-tuning has an essential limitation that some datasets cannot be memorized even with an infinite number of trainable tokens, which can lead to severe underfitting on datasets that cannot be resolved by merely increasing the number of trainable parameters.
>
> We also emphasize that the role of memorization here is in the context of theoretically characterizing the representative power of neural networks and is not directly associated with overfitting, which results in poor accuracy numbers or task specific metrics.
>
> **Regarding scenarios where LoRA is better than prompt-tuning**
>
> A concrete example of when prompt-tuning needs more trainable parameters is Theorem 2 where prompt-tuning cannot memorize the constructed dataset with even an infinite number of trainable tokens.
>
> For more general cases, in eq(16) $m_p \geq n$ does not guarantee that $rank(A) =n$, which is dependent on the property of the dataset $X_1, …, X_n$. If $rank(A) < n$ for prompt length $m_p = O(n)$, prompt-tuning requires more trainable parameters than LoRA. ( eq(16) in the appendix should be $rank(W_v P A^T) = n$ due to a typo in the submitted version, which will be corrected in the updated manuscript)
>
> More practically speaking, since the representative power of prompt tuning is worse than LORA, prompt tuning can only capture functions that deviated not too much from the given transformer. Therefore, when the target task is closer to the pretraining task, prompt tuning would be suitable; while when the target task is drastically different from the pretraining task, prompt tuning may not be able to capture the function and thus LORA could be a better choice.
>
> **Some assumptions are given and not sure whether they are applied in real-world applications**
>
> We made two assumptions in Section 5. Assumption 1 ensures that our analysis is conducted under a non-trivial setting where prompt-tuning and LoRA won’t lose their expressive power trivially.  Assumption 2 is required for the limitation result in Theorem 2. As we have discussed in Lemma 4 and Example 1, this assumption is mild and is supported by some related empirical works.
>
> **Typos**
>
> We will fix the typos in the updated version of the manuscript.
>
> If the response satisfies your concerns, we hope that you will improve the score to reflect the same. We are happy to answer any more questions.

---

> > ### Comment · Reviewer_AVty · 2023-08-17
> >
> > Thank you for your thorough and comprehensive response to my feedback and concerns. I appreciate the effort you've put into addressing each of the points I raised.
> >
> > I'm satisfied with the responses you've provided for the issues I initially brought up. The clarifications regarding the memorization capacity of neural networks, the scenarios favoring LoRA over prompt-tuning, and the applicability of assumptions in real-world applications have indeed addressed my concerns effectively.
> >
> > I am happy to raise my rating. Considering other reviewers' comments, I still encourage authors to add a discussion on the potential directions to improve prompt methods based on the provided analysis.

---

> > > ### Author Response · Authors · 2023-08-18
> > > **Thank you for the updated review**
> > >
> > > Thank you for your updated review and valuable suggestions on the discussion about potential directions.
> > >
> > > We will add a discussion section in our later version to include our thoughts about how to improve prompt methods based on the theoretical analysis (An extension of part of our response to Reviewer VZsv).

---

### Official Review · Reviewer_adez · 2023-07-06

**Soundness:** 3 good
**Presentation:** 2 fair
**Contribution:** 2 fair
**Rating:** 5
**Confidence:** 2

**Summary:**

This paper wants to provide the insights on "when and how to perform prompt tuning to adapt a pretrained transformer to downstream tasks", by theoretically quantifying the universality (i.e., universal approximators) and limitations (i.e., representation capacity) of prompt tuning, in complementing a lot of work focusing on empirically improve soft prompt tuning, such as better initialization or hyper-parameter tuning. Crucially, they show prompt tuning with theoretically very strong transformer models can approximate any seq2seq functions in the set of Lipschitz functions, and they theoretically and empirically show some limitations on weaker transformers (constructed seq2seq datasets as low bounds) by their relatively lower memorization ability.

**Strengths:**

1. This work shows the universality of prompts to the Lipschitz function space, and the limitations of prompt tuning at their memorization ability. The proofs are interesting and well-guaranteed.
2. The mathematical derivations do show your deliberate thinking and rigorous reasoning.
3. The paper writing is almost well-structured.

**Weaknesses:**

1. Just for curious, you define F_L by directly posting the L-lipschitz functions. However, indeed, this causes some unavoidable confusions to readers, as what are "L-lipschitz functions"? Can you provide any context? Especially for some highly theoretical papers?
2. Some writings make me confusing, even though the general structure looks good.
3. Just one question: Is the first section of universality looks very similar to your cited ICLR work on Transformer's universal approximators?

I am happy to raise scores if you can make this more clear to me, such that this highly theoretical paper can obtain a more rigorous consideration for its ratings.

**Questions:**

See weaknesses.

**Limitations:**

See weaknesses.

---

> ### Author Rebuttal · Authors · 2023-08-10
>
> We sincerely thank Reviewer adez for the feedback and valuable suggestions. We have addressed each of the concerns below:
>
> **Regarding explanation on L-Lipschitz function**
>
> In this work, we consider a sequence-to-sequence function $f$ with input $X \in [0, 1]^{d \times m}$. $f$ is an L-Lipschitz function under norm $p$ if $\|f(X_1) - f(X_2)\|_p \leq L \|X_1 - X_2\|_p$.
> Broadly speaking, the Lipschitz property allows the following: give some function $f$ and two distinct inputs $x_1, x_2$, we can bound the difference between the outputs $f(x_1), f(x_2)$ (under some metric) via the difference between the inputs (under a potentially different metric), along with a constant which is the Lipschitz constant. We will provide a more formal definition in the updated version of the manuscript.
>
> Intuitively, this property means the function is continuous in the sense that two close-by inputs are mapped to two close-by outputs, which is a commonly used assumption in theoretical analysis. This property ensures that the approximation error is bounded when we quantize the input/output of seq2seq functions under consideration in Lemma 1.
>
> **Regarding the novelty of Theorem 1**
>
> Yes, in the proof of Theorem 1, part of Lemma 2 and Lemma 3 are borrowed from Yun et. al. However, as discussed previously, from a technical viewpoint Lemma 1 is non-trivial and reduces the construction of a transformer network for universal prompt-tuning to the construction of a transformer network for a meta sequence-to-sequence function, which allows us to leverage some lemmas from Yun et al. Furthermore, we would like to emphasize that we are the first ones to construct a transformer for universal prompt-tuning in the continuous setting. This opens up a broad arena for future work.
>
> If the response satisfies your concerns, we hope that you will improve the score to reflect the same. We are happy to answer any more questions.

---

### Official Review · Reviewer_E8G5 · 2023-07-09

**Soundness:** 3 good
**Presentation:** 3 good
**Contribution:** 3 good
**Rating:** 5
**Confidence:** 3

**Summary:**

This paper focuses on understanding the role and theoretical underpinnings of soft-prompt tuning in transformer-based architectures, which is a technique used to adapt pretrained language models for new tasks. Despite the empirical effectiveness of this method, there's a lack of theoretical understanding of how it differs from tuning the model weights.

The paper takes significant steps in several areas:

1. **Universal Approximation Analysis:** The researchers prove a universality result that guarantees the existence of a transformer with a prompt that can approximate any sequence-to-sequence function within the set of Lipschitz functions, which are a class of functions that have a particular property of bounded rate of change.

2. **Limitations of Limited-Depth Transformers:** The researchers construct datasets that cannot be memorized by a prompt of any length for a single-layer transformer, demonstrating the inherent limitations of prompt-tuning in this setting. The paper extends the analysis to multi-layer transformer settings, providing conditions under which transformers can only learn from invertible functions. This shows additional limitations of the transformer model.

3. **Lower Bound on Tunable Prompt Parameters:** They establish a lower bound on the required number of tunable prompt parameters and compare this with the parameters needed for a low-rank update (LoRA). This helps to illuminate the resource trade-offs between these two methods.

4. **Empirical Results:** The researchers also provide empirical results that back up their theoretical findings, adding credibility to their conclusions.


**Strengths:**

One of the strengths of the paper is its theoretical grounding. It provides a theoretical understanding of the difference between 'tuning parameters before the input' and 'the tuning of model weights'. The paper presents a strong universality result, stating the existence of a strong transformer with a prompt that can approximate any sequence-to-sequence function in the set of Lipschitz functions.

In addition, the paper presents a detailed analysis of the limitations of prompt-tuning, especially for limited-depth transformers. The authors create specific datasets that cannot be memorized by any prompt of any length for a single encoder layer. This analysis provides useful insights into the constraints of prompt-tuning.

The paper is generally easy-to-follow in idea. Although I do not check the theoretical correctness of these proofs.

**Weaknesses:**


While this paper takes a commendable effort in delving into the theoretical underpinnings of prompt-tuning in transformer-based architectures, it has several significant shortcomings.

Firstly, the focus on sequence-to-sequence encoder models poses a substantial gap in the relevance of these findings to practical, real-world application. The universality of sequence-to-sequence transformers cannot be immediately generalized to decoder models, which are commonly used in practice and can handle an arbitrary number of output tokens. Therefore, the study needs to address how its theoretical results can be extended to these types of models. Without this, the scope of the paper's relevance becomes significantly narrow and its conclusions may not be directly applicable to most use-cases.

Secondly, the reliance on the universality results from Yun et al.'s work ("Are transformers universal approximators of sequence-to-sequence functions?") reduces the novelty of the paper. While it is valid to build upon previous work, the authors have not demonstrated a significant extension or new application of these universality findings. It would be beneficial for the paper to provide more novel insights or extend the known results to a more general setting beyond what is already known from the referenced work.

Lastly, the exploration of limitations in prompt-tuning is an interesting aspect of the paper. However, the limited breadth of the experiments conducted undermines the impact of these findings. The limitation proofs revolve around specific, constructed single-layer transformers, which do not cover the wide array of transformer architectures used in practice. The implications of these limitations would be far more convincing if the authors could generalize the findings to multi-layer transformers or a wider variety of architectures.

To improve upon these shortcomings, the authors might consider broadening their analysis to include decoder models, providing more novel extensions to the universality results, and expanding the scope of their limitation analysis to apply to a broader range of transformer architectures.

**Questions:**

1. How to extend the conclusion to decoder models like GPTs that can generate new content?

2. Can you provide some specific examples (with detailed values) for the limitation analysis? E.g. assume the dimension $d = 1$, then x and y are all just numbers. Then I can have a better understanding of why some functions  are not learnable via prompt tuning.

I'm not familiar with theoretical analysis, and skipped the proofs. I will leave the judgement of proofs to my peer reviewers and area chairs.

In addition, several typos should be fixed:

"We use the following notations throughout the paper. A bold lower case character, eg x,": eg --> e.g.

"If yes, can we construct the a transformer for this universality result?": the a --> a



**Limitations:**

Not fully addressed. The authors do not mention the limitation in encoder-only models.

---

> ### Author Rebuttal · Authors · 2023-08-10
>
> We sincerely thank Reviewer E8G5 for the feedback and valuable suggestions. We have addressed each of the concerns below:
>
> **Regarding the extension of Theorem 1 to decoder models**
>
> Thank you for your valuable suggestion. The universality result in Theorem 1 can be extended to decoder models with some minor modifications to the transformer network construction.
>
> For a decoder model, the output is generated by recursively feeding it with the input and output that is already generated. Therefore, the input length can be changing throughout the generation process.
>
> To generalize Theorem 1 to decoder models, we consider a set of L-Lipschitz sequence-to-sequence(seq2seq) functions $F_{L, M}$, where functions in $F_{L, M} $ accept an input sequence $X \in [0, 1]^{d \times m}$, $1 \leq m \leq M$ and $M$ is the maximum input/output length.
> We can prove that a transformer network $\tau$ can be constructed such that for any $f \in F_{L, M}, X\in[0, 1]^{d \times m}, m \leq M$ and $\epsilon > 0$, we can find a prompt $P \in [0, 1]^{d \times m_p}$ such that $d(\tau([P, X])_{:, m_p:}, f(X)) \leq \epsilon$.
>
> The key idea is that the same attention layers used in Lemma 2 proof can generate unique contextual mappings for inputs with arbitrary lengths $m \leq M$. Therefore, we only need to increase the capacity of the final series of MLP layers to map the additional contextual mappings for $m \lt M$.
>
> By constructing the final series of MLP layers, $\tau$ can approximate the desired output token for any input sequence at each generation iteration with an appropriate prompt $P$, which can therefore approximate any seq2seq function under the setting of decoder models.
>
> We will add this extension and detailed proof in our revision of the manuscript.
>
> **Regarding the novelty of Theorem 1**
>
> Although proof of Theorem 1 is partially dependent on some lemmas (Part of Lemma 2 and Lemma 3) from Yun et al, from a technical viewpoint Lemma 1 is non-trivial and reduces the construction of a transformer network for universal prompt-tuning to the construction of a transformer network for a meta sequence-to-sequence function, which allows us to leverage some lemmas from Yun et al. Furthermore, we would like to emphasize that we are the first ones to construct a transformer for universal prompt-tuning in the continuous setting. This opens up a broad arena for future work.
>
> **Regarding the implications of limitation results**
>
> Our limitation results in Theorem 2 and 3 apply to standard single-layer transformers under mild assumptions (Assumption 2 and 3), which are building blocks of a variety of transformer networks. Therefore, studying the limitation of single-layer transformer layers provides a lot of insights for general transformer networks. For analysis on multi-layer transformers, as the reviewer has pointed out already in the paper summary, we have extended to multi-layer transformers in Theorem 4, which provides conditions under which these general purpose transformers learn a very small class of functions and are thus weak.
>
> **Regarding examples with $d=1$**
>
> As we have discussed in Assumption 2, which is a prerequisite assumption for Theorem 2, the dimension of a token $d$ should satisfy $d \geq 2 + \text{dim}((\text{MLP}^{-1}(y_{10}) - x_0) \cup (\text{MLP}^{-1}(y_{20} )- x_0))$ where $\text{dim}(Q)$ measures the dimension of a subspace spanned by vectors in the set $Q$. Therefore, $d$ should be at least 3 in our analysis, which is also a reasonable assumption in practical seq2seq datasets.
>
> **Typos**
>
> We will fix the typos in the updated version of the manuscript.
>
> If the response satisfies your concerns, we hope that you will improve the score to reflect the same. We are happy to answer any more questions.

---

> > ### Comment · Reviewer_E8G5 · 2023-08-15
> >
> > Can you provide some specific examples (with detailed values) for the limitation analysis for $d=3$?

---

> > > ### Author Response · Authors · 2023-08-16
> > > **Follow-up**
> > >
> > > Thank you for the follow-up question.
> > >
> > > Here we provide an example with $d=3$ - for Example 1 in main paper but with simpler weights.
> > >
> > > Assume $W_1 = W_2 = 0$, then $MLP^{-1}(y_{10}) = \{ y_{10} \}, MLP^{-1}(y_{20}) = \{y_{20}\}$.
> > > We can choose $y_{10} = [3.0, 0.0, 0.0], y_{20} = [2.0, 0.0, 0.0], x_0 = [1.0, 0.0, 0.0]$.
> > > Then we have $\texttt{dim}((MLP^{-1}(y_{10}) - x_0)\cup (MLP^{-1}(y_{20}) - x_0)) = \texttt{dim}(\{[2.0, 0, 0], [1.0, 0, 0]\}) = 1$.
> > >
> > > Therefore, to memorize the dataset $\{(X_1 = [x_1, x_0], [y_{11}, y_{10}]), (X_2 = [x_2, x_0], [y_{21}, y_{20}])\}$, we have $Att(x_0, [P, X_1]) = [2.0, 0.0, 0.0], Att(x_0, [P, X_2]) = [1.0, 0.0, 0.0]$.
> > >
> > > Then we can choose $c_1 = [0.0, 1.0, 0.0], c_2 = [0.0, 0.0, 1.0]$ and construct $x_1, x_2$ such that $Att(x_0, [x_1, x_0]) = a_1 c_1$, $Att(x_0, [x_2, x_0]) =a_2 c_2$ where $a_1, a_2$ are non-zero scalars.
> > >
> > > $x_1$ and $x_2$ can be solvable according to Lemma 7, but the exact values are dependent on different attention weights.
> > >
> > > Here we give an example for $Att(x_0, [x_1, x_0])$ and assume that $W_v, W_k, W_v$ are identity matrices. Then we need to solve $x_1$ such that
> > > $$
> > > Att(x_0, [x_1, x_0]) = \frac{\exp{x^\top_0 x_1}}{\exp{x^\top_0 x_1} + \exp{x^\top_0 x_0}} x_1 + \frac{\exp{x^\top_0 x_0}}{\exp{x^\top_0 x_1} + \exp{x^\top_0 x_0}} x_0 \parallel [0.0, 1.0, 0.0]
> > > $$
> > > We set $x_1 = [-k, k, 0.0]$. When $k \to -\infty$, $Att(x_0, [x_1, x_0]) \to x_1$. When $k \to \infty$, $Att(x_0, [x_1, x_0]) \to x_0$. As $Att(x_0, [x_1, x_0])$ is continuous w.r.t. changing $k$, there must be a $k$ such that $Att(x_0, [x_1, x_0]) \parallel [0.0, 1.0, 0.0]$.
> > >
> > > Then according to eq (8), we have
> > > $$
> > > Att(x_0, P) = \frac{1}{\lambda_{12}}(Att(x_0, [P, X_1]) - \lambda_{11}Att(x_0, [x_1, x_0]))
> > > $$
> > > $$
> > > Att(x_0, P) = \frac{1}{\lambda_{22}}(Att(x_0, [P, X_1]) - \lambda_{21}Att(x_0, [x_1, x_0]))
> > > $$
> > > where $\lambda$ s are non-negative scalars.
> > >
> > > Then we have
> > > $$
> > > Att(x_0, P) = \frac{1}{\lambda_{12}}[2.0, 0.0, 0.0] - \frac{\lambda_{11}a_1}{\lambda_{12}}[0.0, 1.0, 0.0]
> > > $$
> > > $$
> > > Att(x_0, P) = \frac{1}{\lambda_{22}}[1.0, 0.0, 0.0] - \frac{\lambda_{21}a_2}{\lambda_{22}}[0.0, 0.0, 1.0]
> > > $$
> > > As both $\frac{\lambda_{11}a_1}{\lambda_{12}}$ and $\frac{\lambda_{21}a_2}{\lambda_{22}}$ are non-zero scalars, there is no solution for $P$ to the above equation set.

---

### Official Review · Reviewer_VZsv · 2023-07-11

**Soundness:** 3 good
**Presentation:** 3 good
**Contribution:** 3 good
**Rating:** 6
**Confidence:** 3

**Summary:**

This paper presents theoretical analysis of prompt tuning by 1) formally define the attention and prompting operations, 2) deriving the universality of prompt tuning with a strong transformer model, 3) further discussing the limitation of prompt tuning under the case of single and multi-layered transformer models. Explorative experiments on standard models and benchmark datasets demonstrate the correctness of the derived theorems.

**Strengths:**

- The paper is well written with clearly defined definitions and theorem presentations.
- Exploring prompt tuning from a theoretical perspective is very interesting and important, since better understanding the limitations of prompts is the key to improve prompting techniques.
- The theorems seems correct and solid.

**Weaknesses:**

- The experimental part is a bit weak compared with the theoretical part.
- It will be nice to have some further discussions on how to leverage the current limitations of prompt tuning to design better prompting-based models.

**Questions:**

Please see weaknesses part.

**Limitations:**

No limitations or potential negative societal impacts discussed in the main context.

---

> ### Author Rebuttal · Authors · 2023-08-10
>
> We sincerely thank Reviewer VZsv for the feedback and valuable suggestions. We address each of the concerns below:
>
> **Regarding additional experiments**
>
> Please see the general response.
>
> **Regarding discussions on potential directions to improve prompt-tuning**
>
> From the analysis in Theorem 2 and 3, we note that the limitation of prompt-tuning primarily arises from the correlation across different inputs. Broadly describing, prompt-tuning implements transformation on different inputs via “additional attention values”, which is more restrictive as compared to the transformations from linear layer/MLP layer on input tokens. An interesting potential direction is: “designing a mechanism to leverage prompting in order to generate prompt-dependent adapter/LoRA updates”.
>
> Since the primary emphasis of the paper is to provide one of the first theoretical works to understand soft prompt tuning, we plan to focus on designing novel prompt-tuning strategies in future work.
>
> If the response satisfies your concerns, we hope that you will improve the score to reflect the same. We are happy to answer any more questions.

---

### Official Review · Reviewer_1J6k · 2023-07-26

**Soundness:** 3 good
**Presentation:** 3 good
**Contribution:** 4 excellent
**Rating:** 7
**Confidence:** 3

**Summary:**

In this work, the authors embark on exploring the capabilities of prompt tuning in the continuous regime,
contrasting it with fine-tuning, as an initial endeavor toward theoretical comprehension. The authors prove
by construction that prompt tuning admits universal approximation within the space of Lipschitz
functions. Additionally, the authors identified inherent limitations of prompt tuning on single-layer transformers by constructing theoretically difficult datasets for prompt tuning. These limitations are then
extended to multi-layer settings under a specific prompt-norm restriction.

**Strengths:**

1. The authors characterize the universal nature of prompt tuning by explicitly constructing a transformer
network.

2. The authors provide a construction-based argument for sequence-to-sequence datasets that cannot be
learned by prompt tuning with a given single-layer transformer.

3. The authors provide the lower bound on the required number of parameters for prompt tuning to
memorize any sequence-to-sequence functions.

4. The authors provide a sufficient condition for multi-layer transformers, under which datasets with
shared output tokens cannot be learned with prompt tuning.

Overall, this paper is well written. The hypothesis of these theories is acceptable and the theoretical analysis is solid.


**Weaknesses:**

Please add more experiments to support the theoretical result.

**Questions:**

N/A

**Limitations:**

the authors adequately addressed the limitations in last page.

---

> ### Author Rebuttal · Authors · 2023-08-10
>
> We sincerely thank Reviewer 1J6k for the positive feedback and valuable suggestions. For the corresponding additional experiments to validate our theorems, please see the general response. If the response satisfies your concerns, we hope that you will improve the score to reflect the same. We are happy to answer any more questions.

---

### Author Rebuttal · Authors · 2023-08-10

We thank all the reviewers for their feedback and valuable suggestions.

**General response on additional experiments**

As requested by most reviewers, we add additional experiments to further validate our theoretical results in the paper.

In Section 7, we designed two experiments to validate our main results for single-layer and multi-layer transformer networks, respectively. In addition to Figure 2, we have another plot for the dataset (WMT14 EnFr) in Appendix Figure 3. Here we did an additional experiment on WMT16 EnDe and showed the spectral norm of soft prompt and loss in the following table.

| Steps                        | 5      | 10     | 15     | 20     | 25     | 30     | 35     | 40     | 45     | 50     |
|------------------------------|--------|--------|--------|--------|--------|--------|--------|--------|--------|--------|
| Spectral Norm of Soft Prompt | 2.4254 | 2.4728 | 2.5173 | 2.5465 | 2.5646 | 2.5772 | 2.5864 | 2.5900 | 2.5910 | 2.5912 |
| Validation Loss              | 0.9478 | 0.9201 | 0.9008 | 0.9062 | 0.8910 | 0.8917 | 0.8915 | 0.8967 | 0.8970 | 0.8959 |


As we can see from Figure 3 and this table, the results are consistent with Figure 2 that there is a clear trend of increasing spectral norm of soft prompt during prompt-tuning, which validates our theoretical analysis in Theorem 4.

For Figure 1, we did an additional experiment to show that LoRA can perfectly memorize the constructed dataset with two examples with LoRA width $r=2$. With LoRA width $r=2$, it can achieve training loss close to 0 (less than 1e-10) which is similar to the performance of full-fine tuning on the constructed dataset. This additional result validates our Theorem 3 that LoRA can memorize a dataset with $n$ examples with trainable parameters $O(n)$ while prompt-tuning may require more.

We will add the additional experiment results and figures in our future manuscript. We hope the reviewers can re-evaluate our work based on these responses.

---

### Comment · Area_Chair_aFmB · 2023-08-14
**Reviewer-author discussion**

Dear Reviewers,

Please take a moment to read the authors' responses. Your insights and feedback are crucial in ensuring a comprehensive evaluation. Thanks.

AC

---

### Decision · Program_Chairs · 2023-09-21

**Decision:**

Accept (poster)

**Comment:**

The reviewers' feedback reflects a positive assessment of the paper's theoretical foundations and contributions. They highlight the paper's exploration of prompt tuning in the continuous regime, backed by theoretical analyses and proofs. The proven universal approximation capability within the Lipschitz functions space is acknowledged as a key strength. Through the reviewers-authors discussion phase, the reviewers express satisfaction with the provided clarifications and improvements, leading to an increased rating. Additionally, the authors are suggested to consider discussing potential directions for enhancing prompt methods based on the presented analysis.